# SAO-Instruct: Free-form Audio Editing using Natural Language Instructions

**Michael Ungersböck**
ETH Zurich
mungersboeck@ethz.ch

**Florian Grötschla**
ETH Zurich
fgroetschla@ethz.ch

**Luca A. Lanzendörfer**
ETH Zurich
lanzendoerfer@ethz.ch

**June Young Yi**
Seoul National University
julianyi1@snu.ac.kr

**Changho Choi**
Korea University
changho98@korea.ac.kr

**Roger Wattenhofer**
ETH Zurich
wattenhofer@ethz.ch

## Abstract

Generative models have made significant progress in synthesizing high-fidelity audio from short textual descriptions. However, editing existing audio using natural language has remained largely underexplored. Current approaches either require the complete description of the edited audio or are constrained to predefined edit instructions that lack flexibility. In this work, we introduce SAO-Instruct, a model based on Stable Audio Open capable of editing audio clips using any free-form natural language instruction. To train our model, we create a dataset of audio editing triplets (input audio, edit instruction, output audio) using Prompt-to-Prompt, DDPM inversion, and a manual editing pipeline. Although partially trained on synthetic data, our model generalizes well to real in-the-wild audio clips and unseen edit instructions. We demonstrate that SAO-Instruct achieves competitive performance on objective metrics and outperforms other audio editing approaches in a subjective listening study. To encourage future research, we release our code and model weights.

## 1 Introduction

Generative audio models have become increasingly popular, allowing users to generate high-fidelity long-form audio within seconds. While they have been successfully applied in various domains, including music [8, 10, 1, 5], speech [29, 26, 21, 4], and general audio [30, 24, 9, 13], the area of audio editing still remains largely unexplored. These models, especially when given short or ambiguous prompts, have freedom and flexibility in their outputs. While this enables diverse and creative generations, it may lead to outputs that deviate from the user's original intent. A similar issue exists in recorded audio, which often contains imperfections and typically requires manual editing before being suitable for real-world use. These modifications can range from subtle adjustments to stylistic and spatial effects, such as *"the birds should chirp louder"*, *"make it sound muffled"*, or *"add reverb to the fireworks."* In addition to the challenges faced in generative audio, such as modeling the high-dimensional long-term temporal dependencies, audio editing introduces further complexities. Edits must selectively modify specific aspects of the provided audio while preserving the overall background context. User-specified edits can also vary dramatically in scope and typically do not follow rigid structures.

Recent methods have made progress towards addressing some of these challenges. Zero-shot inversion approaches [33, 20] modify existing audio by conditioning on a full target description. However, describing an audio clip with its unique sound characteristics in concise, unambiguous text is challenging and requires significant effort [1]. A more intuitive alternative is instruction-based

39th Conference on Neural Information Processing Systems (NeurIPS 2025).

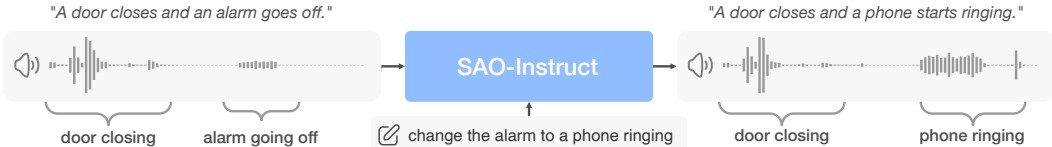

Figure 1: Given an input audio clip and an edit instruction, SAO-Instruct outputs the edited audio while keeping the overall audio context intact.

editing, where users specify only the intended change in natural language rather than the desired outcome. The AUDIT [44] model is one of the first to explore instruction-based editing, but only supports a predefined set of editing tasks. Additionally, user instructions are often diverse and underspecified, which prevents them from aligning neatly with such fixed operations. We argue that enabling **fully free-form instruction-based editing** would allow more expressive transformations, significantly simplify user interaction, and broaden the applicability of generative audio models.

In this work, we introduce SAO-Instruct, the first model for free-form instruction-based editing in the audio domain. We use Prompt-to-Prompt [16], DDPM inversion [33], and a manual editing pipeline to generate data triplets of input audio, edit instruction, and the corresponding edited audio. As illustrated in Fig. 1, we show that the model learns to modify audio given a free-form edit instruction. While partially fine-tuned on synthetic data, our experiments show that the model is able to generalize well to real in-the-wild audio and is able to follow diverse editing operations.

Our key contributions can be summarized as follows:

- We present SAO-Instruct, the first fully free-form instruction-based audio editing model based on Stable Audio Open. Our model generalizes well to real in-the-wild audio clips and achieves competitive performance compared to zero-shot approaches that rely on full audio descriptions.
- We design a novel pipeline to create a dataset of audio editing triplets (input audio, edit instruction, output audio), combining LLM-based prompt generation, Bayesian Optimization, and a filtering mechanism that addresses the limitations of current generative audio models.
- We create a diverse dataset using three complementary approaches: fully synthetic generation via Prompt-to-Prompt, semi-synthetic samples via DDPM inversion, and real-world manual edits. We demonstrate their contribution to the performance of our editing model in an ablation study.

**Samples are available online:** `https://eth-disco.github.io/sao-instruct`

## 2 Related Work

### 2.1 Text-to-Audio Generation

The area of generative audio has gained significant popularity in recent years. Applications range from music generation [8, 10, 1, 5] and speech synthesis [29, 26, 21, 4] to the broader general audio domain [30, 24, 9, 13]. The latter category encompasses everything from short, distinct sound effects (*"the sound of a whip"*) to more complex scenes (*"frogs croaking at a pond"*) and environmental ambient noise (*"light rain with distant thunder"*). Several architectures have been explored for general audio synthesis. Apart from autoregressive approaches [24], recent advancements in latent diffusion models have achieved state-of-the-art results. AudioLDM [30] uses a latent diffusion architecture conditioned on Contrastive Language-Audio Pretraining (CLAP) [45] embeddings to learn the latent space of mel spectrograms. While such approaches require a vocoder to reconstruct the waveform, which can introduce artifacts due to missing phase information, recent advances such as BigVGAN [28] have significantly improved reconstruction fidelity. To remove the need for vocoders, Stable Audio Open [9] uses a variational autoencoder operating at a 21.5 Hz latent framerate to encode stereo audio into a continuous latent representation. A diffusion transformer [38] then operates on this latent space, conditioned on the text prompt, timing information, and the current diffusion timestep. Finally, the output is decoded back into 44.1 kHz stereo audio of up to 47 seconds in length. By conditioning on timing information, users can specify the start and end points of the generated audio, while the model is trained to fill the remaining segments with silence.

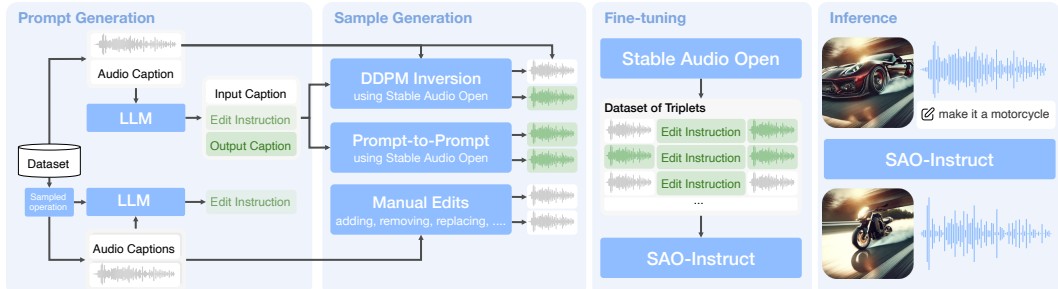

Figure 2: Overview of our proposed method. Green indicates synthetic data. Audio datasets are used as the starting point for prompt generation. DDPM inversion and Prompt-to-Prompt use the input caption and generated output caption to create a partial and fully synthetic dataset, respectively. For manual edits, a predefined edit operation is sampled. In the fine-tuning stage, Stable Audio Open is trained on the combined generated samples and edit instructions. During inference, SAO-Instruct receives an audio clip and a free-form edit instruction and produces the edited output.

## 2.2   Generative Editing

Recent progress in generative models has enabled new capabilities in editing content across various domains. Approaches to generative editing can be broadly categorized into zero-shot methods, which adapt existing models without training, and supervised methods, which fine-tune generative models for specific editing tasks.

**Zero-shot Methods.**   A common zero-shot editing strategy for diffusion models involves partially noising an input sample and then guiding the denoising process with a modified description [36]. Other approaches [37, 19, 33, 20] use inversion techniques which first estimate the latent representation given an input sample and its description, and then generate the edited output by guiding the denoising process with a modified prompt. However, these inversion approaches typically require explicit access to detailed input and output descriptions, which are often unavailable in real-world settings. The quality of results is also highly dependent on how these descriptions are phrased, making such approaches less intuitive for practical use.

**Supervised Methods.**   Another line of work trains generative models specifically for editing tasks. Diffusion models with an infilling objective learn to reconstruct masked audio regions based on the surrounding context and a target text description [43]. A more intuitive alternative is instruction-based editing, where users specify only the intended change in natural language rather than the desired outcome. In the image domain, InstructPix2Pix [3] fine-tunes a diffusion model on synthetic data triplets (input image, edit instruction, output image), generated using an LLM and Prompt-to-Prompt [16], which enables image edits by selectively injecting attention maps of the original prompt during the generation of the modified prompt. In the general audio domain, AUDIT [44] was among the first to enable instruction-based editing. It supports the following five core operations: addition, removal, replacement, inpainting, and super-resolution of audio. InstructME [14] introduced a related approach for music editing that also considers the harmonic consistency while executing edits. Fugatto [42] further explores free-form audio generation and transformation through specialized multitasks datasets. While such models demonstrate broader instruction-following capabilities, they are not specifically tailored for high-fidelity audio editing using natural free-form instructions.

## 3   Method

Free-form audio editing with a supervised learning approach requires a dataset of triplets consisting of an input audio clip, an edit instruction, and the corresponding edited audio. Since no such dataset is readily available, we construct one using a combination of LLM-based prompt generation and partially synthetic audio, inspired by recent work in the image domain [3]. As illustrated in Fig. 2, our approach first prepares a dataset of prompts that contain an input caption, an edit instruction, and an output caption. The corresponding paired audio samples are created using three complementary methods: fully synthetic audio generated via Prompt-to-Prompt [16], semi-synthetic audio via DDPM

inversion [33], and manually edited real audio clips. These generated edit instructions and audio pairs are then combined to form the full training set. Finally, Stable Audio Open (SAO) [9] is fine-tuned on this dataset to obtain SAO-Instruct, which can edit a given audio clip based on a free-form natural language instruction.

## 3.1 Prompt Generation

To train our model for free-form editing, we need samples of triplets (input caption, edit instruction, and output caption). For the input captions, we use the captioning datasets AudioCaps [22] and WavCaps [35]. AudioCaps contains 50k human-written captions paired with audio clips sourced from AudioSet [12]. WavCaps consists of 400k audio-caption pairs collected from multiple sources, including FreeSound [1], AudioSet-SL [15] and the BBC Sound Effects [2] library. We use GPT-4o to generate a synthetic dataset of structured prompts. Given an input caption, the LLM is prompted to generate a fitting edit instruction and a corresponding output caption. For example, starting from the caption *"Birds chirping and water flowing"*, it may generate the instruction *"Remove the water flowing"* and the output caption *"Birds chirping"*, with the output caption derived by applying the edit instruction to the input caption. The LLM also generates additional metadata, including negative prompts and a count of distinct audible elements, which we use to improve sample quality for audio synthesis and to enable downstream filtering. More details can be found in Appendix A.

## 3.2 Prompt-to-Prompt

A significant challenge in audio editing is applying targeted modifications while preserving the overall audio context, including background sounds and overall atmosphere. Since there is no such dataset available, we adapt the Prompt-to-Prompt [16] method, originally developed for image editing, to the audio domain. Prompt-to-Prompt enables localized edits of synthesized audio by injecting attention maps from the input prompt into the generation process of the edited prompt. We use Stable Audio Open [9] as the underlying generative model, due to its high-fidelity 44.1 kHz stereo output and flexibility of generating audio up to 47 seconds in length. As Stable Audio Open sometimes omits sounds present in captions with connectors, such as *"Helicopter taking off with wind blowing and dogs barking"*, we fine-tune it on AudioCaps to improve prompt adherence on general audio. While fine-tuning improved the alignment between prompts and outputs, as detailed in Appendix D, achieving more consistent high-quality results also requires selecting a suitable combination of seed and Classifier-Free Guidance (CFG) [18] values. To this end, our approach consists of two stages, as shown in Fig. 3. First, our method explores different combinations of seed and CFG values to identify a configuration that produces satisfactory results for a given input/output caption pair. Next, Prompt-to-Prompt is applied to generate the edited audio pair.

**Candidate Search.** The process for finding a suitable candidate configuration (seed and CFG value) for each prompt pair is illustrated in Fig. 3, section (a) Candidate Search. We use Stable Audio Open to generate seven audio pairs, each with a different configuration. Specifically, each pair is generated using 50 denoising steps, a randomly chosen seed, and a CFG value sampled between 3 and 9. To assess the similarity between the generated audio and corresponding caption, we use the Gemini 2.0 Flash API. We instruct Gemini to perform a perceptual quality check by analyzing all audible elements in the provided audio and assigning a score between 1 and 10, based on how closely the generated audio resembles the provided caption. Samples that score above a threshold of 6 proceed to the next stage, in which the CLAP similarity is calculated. Finally, we select the configuration that passes the Gemini filter and has the highest mean CLAP similarity across input and output.

**Sample Generation.** After finding a suitable configuration for a prompt, we move on to the sample generation as outlined in section (b) of Fig. 3. There are various parameters that can be configured in Prompt-to-Prompt. The *injection fraction* $\lambda_{\text{frac}}^{\text{attn}}$ configures the extent to which attention maps from the input audio influence the generation of the output audio. A fraction of 0 means that no attention maps from the input audio are injected. As a result, the output is not directly influenced by the input, apart from shared initial noise and sampling noise. In contrast, an injection fraction of 1 enforces strong similarity such that the output audio closely resembles the input audio. Intermediate values allow

---

[1] https://freesound.org/
[2] https://sound-effects.bbcrewind.co.uk/

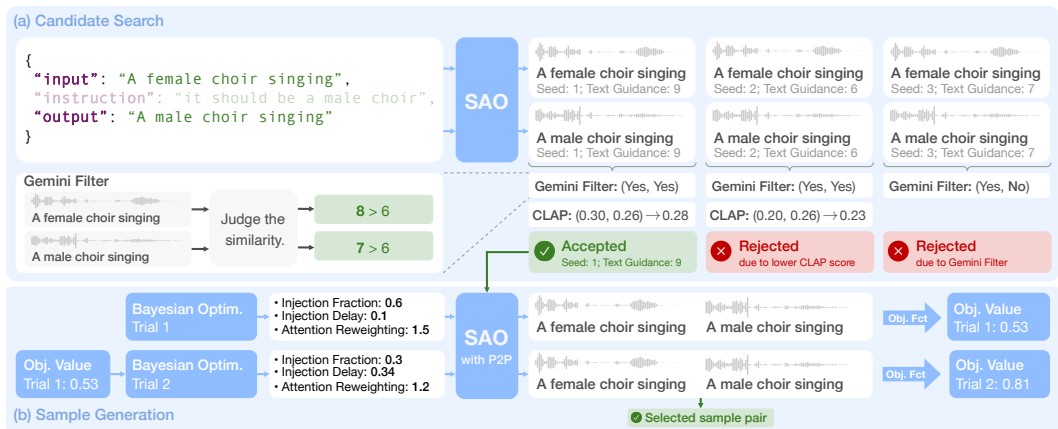

Figure 3: Pipeline for Prompt-to-Prompt audio generation. In (a), various seeds and CFG value combinations are tested and filtered using Gemini and CLAP to identify suitable configurations for prompts. In (b) Stable Audio Open (SAO) with Prompt-to-Prompt generates audio pairs using the seed and CFG configuration found in (a). A Bayesian Optimization process suggests Prompt-to-Prompt parameters and resulting samples are evaluated using an objective function. For clarity, only 3 candidate pairs and 2 Bayesian Optimization trials are shown.

for a balance between flexibility and audio similarity during generation. The *injection delay* $\lambda_{\text{delay}}^{\text{attn}}$ controls at which point the attention injection begins, expressed as a fraction of the total number of attention maps. A value of 0 indicates that the injection starts from the first attention map, while higher values delay the injection. For example, a value of 0.3 skips the first $30\%$ of attention maps before the injection begins. As a constraint, it follows that $\lambda_{\text{frac}}^{\text{attn}} + \lambda_{\text{delay}}^{\text{attn}} \leq 1$. *Attention reweighting* $\lambda_{\text{weight}}^{\text{attn}}$ specifies a multiplier that increases attention to tokens that differ between the input and output captions. These tokens are identified by comparing the tokenized versions of both captions, and their corresponding cross-attention maps are scaled by the given factor. For example, a value of 1.5 increases the attention on changed tokens or newly added tokens by 50%, which helps the model pay closer attention to proposed edits.

Using a fixed Prompt-to-Prompt configuration often leads to suboptimal results. While some instructions involve only subtle changes (e.g., *"add an echo effect"*), others require more extensive modifications (e.g., *"remove the people talking"*). To handle this variability, we assign a unique parameter configuration to each prompt pair. Since manual tuning is infeasible at scale, we rely on Bayesian Optimization [2] to automatically select suitable parameters. This setup requires an objective function that evaluates both the quality of the generated audio pair and the edit accuracy.

**Objective Function.** We define an objective function that evaluates the quality of the generated audio pair and how well the edit was executed, using a combination of multiple metrics. Building on insights from prior work, particularly InstructPix2Pix [3], as well as our own experiments, we identified the following metrics as suitable. The output CLAP similarity $M_{\text{CLAP}}^{\text{out}}$ measures the cosine similarity between the CLAP embeddings of the generated output audio and the output caption. $M_{\text{CLAP}}^{\text{dir}}$ measures the direction of change in the embedded CLAP space, as initially introduced for the image domain [11]. It calculates the cosine similarity between the difference of the CLAP embedded audio pair and the CLAP embedded prompt pair. The CLAP audio similarity $M_{\text{CLAP}}^{\text{sim}}$ measures the cosine similarity between the CLAP embedding of the input and output audio. The mel spectrogram audio similarity $M_{\text{MEL}}^{\text{sim}}$ calculates the multi-scale mel spectrogram loss [25] between the generated audio pair. The objective is computed as a weighted sum and is maximized during Bayesian Optimization. Note that the multi-scale mel spectrogram loss is subtracted as lower values indicate better perceptual alignment.

$$\mathcal{L}_{\text{obj}} = \omega_1 \cdot M_{\text{CLAP}}^{\text{out}} + \omega_2 \cdot M_{\text{CLAP}}^{\text{dir}} + \omega_3 \cdot M_{\text{CLAP}}^{\text{sim}} - \omega_4 \cdot M_{\text{MEL}}^{\text{sim}} \tag{1}$$

Since not all metrics contribute equally, we conducted a small-scale listening study to determine effective weightings. Using our pipeline, we generated around 100 audio pairs with different weightings used for the objective function. Listeners were presented with an input audio clip, the edit

instruction, and two output audio samples selected by the objective functions initialized with different weightings. An ELO rating system was used to rank the different weighting configurations based on their effectiveness in selecting high-quality, relevant edits. We found the following weightings to be effective: $\omega_1 = 8$, $\omega_2 = 14$, $\omega_3 = 0.5$, $\omega_4 = 1.5$.

**Optimization.** We run the Bayesian Optimization process for 10 trials for each caption pair. Specifically, we optimize $\lambda_{\text{frac}}^{\text{attn}} \in [0.3, 0.9]$, $\lambda_{\text{delay}}^{\text{attn}} \in [0.0, 0.6]$, and $\lambda_{\text{weight}}^{\text{attn}} \in [1.0, 1.8]$. Stable Audio Open is configured to use 50 denoising steps during each trial, while the final selected audio pair is generated using 100 denoising steps. Using fewer denoising steps during optimization significantly reduces runtime without affecting the relative ranking of generated samples, while the final 100 steps maximize audio fidelity once the optimal parameters are found.

### 3.3 DDPM Inversion

In addition to the fully synthetic Prompt-to-Prompt dataset, we create a semi-synthetic dataset of audio pairs using DDPM inversion to increase data diversity. The main benefit of this approach is that the input audio is a real audio clip, while only the output audio is generated. We use the zero-shot text-based audio (ZETA) [33] implementation of DDPM inversion with Stable Audio Open as the underlying generative model. We provide the model with an existing input audio and caption, with the output caption generated as outlined in Section 3.1.

**Sample Generation.** To use ZETA in an automated manner, several parameters need to be configured per sample. The $\text{CFG}_{\text{src}}$ and $\text{CFG}_{\text{tar}}$ parameters control the CFG strength of the input and output prompt, respectively. Unlike traditional inversion techniques, which are applied to the full denoising process, ZETA uses the parameter $T_{\text{start}}$ to specify up to which timestep the inversion is performed. Lower values for $T_{\text{start}}$ result in high consistency with the input audio, while higher values enable more editing flexibility. As the required edit strength varies across instructions, a fixed value would lead to under- or over-editing. We therefore apply Bayesian Optimization with the objective function as defined in Section 3.2.

**Optimization.** For each caption pair, we run Bayesian Optimization for 7 trials. We search for optimal values within the following ranges: $\text{CFG}_{\text{src}} \in [1, 3]$, $\text{CFG}_{\text{tar}} \in [3, 10]$, and $T_{\text{start}} \in [18, 65]$. Throughout optimization, Stable Audio Open is configured to use 70 denoising steps. Compared to Prompt-to-Prompt, we use fewer optimization trials and denoising steps to balance quality and runtime, given the higher computational cost of DDPM inversion.

### 3.4 Manual Edits

We further extend our dataset using a suite of twelve manually implemented operations, inspired by the approach from AUDIT [44]. We refer to these operations as *manual edits*, as each edit is implemented using standard, deterministic, and interpretable audio effects. The twelve editing tasks are ADD, REPLACE, DROP, SWAP, LOOP, PITCH, SPEED, LOW_PASS, HIGH_PASS, INPAINT, SUPER_RES, and DENOISE. Each task has a set number of inputs, certain constraints on those inputs, and an optional controllable parameter. The input audio clips may be constrained by the number of unique audio elements, as analyzed in Section 3.1, or by their duration. Some tasks accept parameters that introduce variability in their edits. For instance, the PITCH task allows control over the amount of pitch shifting, while the LOOP task accepts how many times the input audio should get repeated. The final output of each operation is an audio pair consisting of the input and edited audio clip. All tasks, input constraints, and parameters are detailed in Appendix B.

**Sample Generation.** To create an audio editing triplet, we first sample one of the twelve tasks with equal probability. Based on the task's constraints, input audio clips are filtered before suitable candidates are randomly selected. After applying the edit operation, we synthesize the natural language instruction by passing the captions of the input audio clips, alongside the optional parameter value, to a task-specific LLM-based instruction generation stage built on GPT-4.1 mini. Each generation process is provided with a description of the task and few-shot examples, allowing the model to produce tailored task-specific instructions. To increase instruction diversity and better reflect how real-world users might phrase requests, we apply two post-processing steps using custom stages based on o4-mini. Each post-processing step is executed with 50% probability, independently. In the

Table 1: An example three-stage instruction generation pipeline for the `ADD` task, where the selected base audio *"People talking in a roadside cafe"* is mixed with the target audio *"A chirping bird"*. First, the initial stage combines the base and target audio captions to produce an edit instruction. Then, the variation stage rewrites it with alternate phrasing to increase diversity. Finally, the minimization stage compresses it into its most concise form. The last two stages are applied with 50% probability.

| Generation Stage | Applied | Instruction |
| --- | --- | --- |
| 1. Initial | 100% | *"Add the sound of a bird chirping to the people talking in a roadside cafe"* |
| 2. Variation | 50% | *"Add some chirping birds to the chatter in a roadside cafe"* |
| 3. Minimization | 50% | *"Add bird sounds"* |

variation stage, the instruction is rewritten without altering its meaning. In the minimization stage, the instruction is compressed into its most concise form. An example of this three-stage pipeline for the `ADD` task is shown in Table 1.

## 4 Experimental Setup

### 4.1 SAO-Instruct

**Dataset.** To generate the dataset of audio editing triplets we use the three approaches outlined in Section 3. For Prompt-to-Prompt, input captions are taken from the AudioCaps [22] training split and a random subset of FreeSound from WavCaps [35]. For DDPM inversion, the input audio and caption are sourced from a random subset of AudioSet-SL [15]. Edit instruction and output captions are generated as described in Section 3.1. For the manual edits, we choose a different subset of FreeSound from WavCaps to avoid overlaps with Prompt-to-Prompt. The final audio samples are stored as 44.1 kHz stereo WAV files. The computational resources used for the dataset generation are listed in Appendix C.

**Fine-tuning and Inference.** We start from the open-source weights of Stable Audio Open and fine-tune the model on our created dataset of triplets: input audio, output audio, and edit instruction. SAO-Instruct uses three types of conditioning: (1) the text prompt is replaced with a free-form edit instruction, (2) the timing condition is set to the length of the provided input audio, (3) an additional audio condition for the input audio is concatenated to the model's input channels. During training, the model learns to modify the provided input audio based on the edit instruction, such that the resulting output matches the reference output audio. During inference, we encode the input audio into the latent space of the diffusion model and add Gaussian noise. This noised latent is used as the initial starting point for the denoising process. Unless specified otherwise, we use 100 denoising steps and a CFG value of 5. More fine-tuning and inference details are found in Appendix D and E.

### 4.2 Evaluation Metrics

**Objective Metrics.** We use several metrics for objective evaluation that capture both distributional similarity and perceptual quality. To measure how closely the edited samples match the distribution of real audio, we compute the Fréchet Distance (FD) [17], the log spectral distance (LSD), and the Kullback-Leibler (KL) divergence. To evaluate perceptual quality, we use the Inception Score (IS) [39]. The FD, KL, and IS metrics utilize the PANNs [23] classifier. The LSD calculates the distance between spectrograms of original and target audio clips. To compute these metrics, we use the evaluation pipeline provided by AudioLDM [30]. Additionally, we evaluate how well edits were performed using the CLAP [45] score, which measures the cosine similarity between the generated audio and the target caption in the CLAP embedding space. As CLAP is also used in our data generation pipeline to filter samples, it can bias results towards its embedding space. We mitigate this by not training SAO-Instruct with a CLAP loss and by reporting multiple objective and subjective metrics.

**Subjective Metrics.** For the subjective evaluation, we conducted a listening study with 13 participants. Each participant was presented with 10 randomly selected 10-second audio clips from the AudioCaps test subset, including the original caption, an edit instruction, and the corresponding

Table 2: Ablation study comparing the influence of different generated datasets on edit relevance and audio quality. FD, LSD, and KL are shown for both original and regenerated audio. Metrics reported with ± indicate mean and standard deviation across evaluation samples. Outputs from SAO-Instruct fine-tuned on the combined dataset balance the strengths of the individual approaches, achieving both accurate edits and faithfulness to the input audio.

| Training Dataset | Samples | Original Ref. | | | Regenerated Ref. | | | IS ↑ | CLAP ↑ |
|---|---|---|---|---|---|---|---|---|---|
| | | FD ↓ | LSD ↓ | KL ↓ | FD ↓ | LSD ↓ | KL ↓ | | |
| Prompt-to-Prompt | 50k | 18.71 | 1.50 | 1.32 | **18.29** | **2.68** | 1.77 | **7.94±0.72** | 0.38±0.14 |
| DDPM Inversion | 50k | 20.50 | **1.34** | 0.86 | 20.72 | 2.75 | 1.87 | 6.82±0.73 | 0.34±0.15 |
| Manual Edits | 50k | **14.60** | 1.42 | **0.58** | 21.21 | 2.76 | 1.89 | 7.50±0.68 | 0.35±0.15 |
| Combined | 50k | 19.11 | 1.41 | 1.02 | 19.24 | 2.72 | **1.74** | 7.69±0.76 | 0.38±0.14 |
| Combined-Large | 150k | 18.38 | 1.36 | 0.93 | 18.97 | 2.72 | 1.76 | 7.59±1.00 | **0.38±0.14** |

edited audio clips generated by the models. Model names were hidden from participants and the order of outputs was randomized on a per-sample basis. We use similar metrics to AudioEditor [20], where participants were tasked to rate each clip using the Mean Opinion Score (MOS) on a discrete scale between 1 and 5. Ratings were collected for three categories: *Quality*, the perceptual quality of the edited audio compared to the original input audio, *Relevance*, how well the edit was performed, and *Faithfulness*, the similarity between the input audio and the edited audio. The input audio refers to the original, unedited clip provided to the model for editing. The final MOS for each category was computed by averaging ratings across all participants and samples. Appendix F shows the evaluation interface and the instructions given to participants.

# 5 Evaluation

We evaluate the performance of SAO-Instruct in an ablation study that measures the impact of the different methods for dataset generation, and compare it with audio editing baselines. We evaluate on 1k 10-second samples from the AudioCaps test subset, where edit instructions and output captions are generated using the approach described in Section 3.1. For the FD, LSD, and KL metrics we use two types of references: the original AudioCaps clips and synthetic audio generated from the target captions using Stable Audio Open. The original reference tries to capture realism with natural audio, while the regenerated reference serves as a proxy to measure how well the edits were performed. Notably, lower FD, LSD, and KL scores do not necessarily reflect better editing performance, as these metrics primarily measure the similarity to the reference audio. For instance, edited audio that is perceptually close to the original audio clips may score well, even if the edit instruction was ignored or only partially followed. By including both types of references, we aim to capture both naturalness and edit accuracy of generated outputs.

## 5.1 Ablation

To identify the effectiveness of the different dataset generation techniques, we fine-tune Stable Audio Open separately on datasets generated using Prompt-to-Prompt, DDPM inversion, and manual edits, each consisting of 50k samples. Additionally, we evaluate two combined variants: one with 50k samples with equal contribution from each of the three generation methods, and a larger version containing 150k samples with 50k samples from each method.

**Results.** The results of this ablation study are shown in Table 2. Edits from SAO-Instruct fine-tuned on Prompt-to-Prompt showcase high similarity with the regenerated reference. This indicates that the performed edits may be more accurate, while being less faithful to the original qualities of the input audio. In contrast, fine-tuning on DDPM inversion and manual edits, which are built on partial or fully real audio, may produce edits that are less precise while better preserving the characteristics and quality of the input audio. Overall, the combined approaches perform robustly across all metrics, balancing the advantages from the individual datasets.

Table 3: Comparison with zero-shot audio editing baselines. Results (mean $\pm$ standard deviation) are shown for applicable objective and all subjective metrics. Inf. denotes the inference time per sample in seconds, measured on a single NVIDIA A6000 GPU with a batch size of 1.

| Models | Inf. (s) $\downarrow$ | Original Ref. | | | Regenerated Ref. | | | IS $\uparrow$ | CLAP $\uparrow$ | Subjective Metrics | | |
| | | FD $\downarrow$ | LSD $\downarrow$ | KL $\downarrow$ | FD $\downarrow$ | LSD $\downarrow$ | KL $\downarrow$ | | | Quality $\uparrow$ | Relevance $\uparrow$ | Faithfulness $\uparrow$ |
|---|---|---|---|---|---|---|---|---|---|---|---|---|
| AudioEditor | 79.49 | **17.21** | 1.73 | 1.40 | 25.97 | **2.32** | 1.46 | **10.01**$\pm$**0.60** | **0.48**$\pm$**0.12** | 3.22$\pm$1.01 | 3.33$\pm$1.35 | 2.75$\pm$0.99 |
| ZETA$_{T_{start}=50}$ | 15.31 | 24.65 | 2.27 | 1.64 | **17.01** | 2.55 | **1.26** | 9.06$\pm$0.67 | 0.38$\pm$0.13 | **3.56**$\pm$**0.83** | 3.25$\pm$1.25 | 2.95$\pm$1.06 |
| ZETA$_{T_{start}=75}$ | 17.78 | 27.91 | 2.69 | 1.86 | 18.88 | 2.59 | 1.36 | 9.07$\pm$0.72 | 0.36$\pm$0.13 | 3.28$\pm$1.00 | 3.04$\pm$1.24 | 2.75$\pm$1.12 |
| SAO-Instruct | **9.94** | 18.38 | **1.36** | **0.93** | 18.97 | 2.72 | 1.76 | 7.59$\pm$1.00 | 0.38$\pm$0.14 | 3.54$\pm$0.93 | **3.83**$\pm$**1.00** | **3.99**$\pm$**0.74** |

## 5.2 Comparison with Baselines

We compare SAO-Instruct with prior zero-shot audio editing baselines. The ZETA baseline [33] uses Stable Audio Open with 100 denoising steps as the underlying generative model, which was trained on FreeSound and the Free Music Archive (FMA) [6]. We evaluate two variants by adjusting the edit strength via the $T_{start}$ parameter, set to 50 and 75. ZETA has access to the full original and target audio description. We also compare our model to AudioEditor [20], which uses Auffusion [46] as its underlying model trained on several audio datasets, including AudioCaps [22], WavCaps [35], and MACS [34]. Depending on the edit type, AudioEditor conditions on either the original or target description, along with the indices of words that changed between the descriptions. SAO-Instruct is trained on a combined dataset of 150k audio editing triplets, consisting of 50k samples each from Prompt-to-Prompt, DDPM inversion, and manual edits. Unlike the baselines, SAO-Instruct only has access to the edit instruction, conveying significantly less information than full audio descriptions.

**Results.** The results of the comparison with baselines are shown in Table 3. SAO-Instruct has the fastest inference time, editing audio in just under 10 seconds, making it nearly 8x faster than AudioEditor. On the FD, LSD, and KL metrics, the models perform similarly, with no model dominating across both reference proxies. AudioEditor has the highest CLAP and Inception scores by significant margin, likely due to its additional information available as the full target audio caption and longer inference time. However, this also leads to more aggressive edits that reduce the similarity with the original audio clip, reflected in its lower faithfulness score. SAO-Instruct shows strong performance in the subjective listening test and outperforms the other models in both edit relevance and faithfulness to the original audio, while maintaining high audio quality. Using only a free-form instruction, SAO-Instruct demonstrates competitive performance and, in several cases, exceeds the results of models conditioned on full audio descriptions.

## 5.3 Limitations

While SAO-Instruct shows promising results, some limitations remain. The generation process for our Prompt-to-Prompt and DDPM inversion datasets is computationally expensive and constrained by the capabilities of the underlying generative model. Examples of failure cases are shown in Appendix G.3. While our work focuses on editing general audio, our approach could be applied to music editing, provided appropriate generative music models exist and triplets are constructed from music datasets. Future work could explore handling multi-step edits and extending support to languages other than English. See Appendix H for a discussion on broader impacts.

## 6 Conclusion

We introduce SAO-Instruct, the first fully free-form instruction-based audio editing model. SAO-Instruct can perform a wide-range of edit instructions, while preserving the overall context and coherence of the provided input audio. We propose a novel data generation pipeline utilizing Prompt-to-Prompt [16], DDPM inversion [33], and manual edits [44] to create a diverse dataset of audio editing triplets. Our evaluations show that SAO-Instruct outperforms existing zero-shot editing baselines in subjective listening tests and achieves competitive performance on objective metrics. While prior baselines require information from both the input and target audio captions to guide the editing process, SAO-Instruct requires only a single free-form edit instruction. We believe free-form instruction-based audio editing unlocks new possibilities for creative audio workflows and hope that our released model weights and code will encourage further research in this area.

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

# A    Prompt Generation

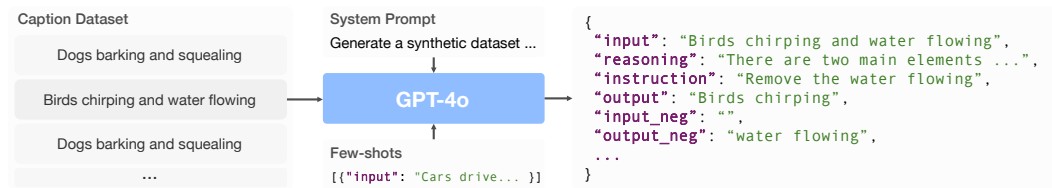

Figure 4: Pipeline for prompt generation. A caption is taken from a dataset and passed to an LLM, which produces an edit instruction and a corresponding output caption. Additional metadata is generated for downstream filtering and for improving sample quality for synthetic audio generation.

**Generation and Filtering.**    As illustrated in Fig. 4, we use GPT-4o to create a dataset of structured prompts. Before generating outputs, the LLM reasons about the provided input caption to identify all audible elements and a suitable edit instruction. The LLM also generates additional metadata, including a count of distinct audio elements of the input caption. Captions vary in complexity, with some containing only a single audible element (*"A cat meows"*), while others may have multiple distinct elements, such as *"A man speaks during heavy traffic with emergency sirens going by"*. Current generative audio models often struggle to accurately synthesize these complex captions, which can result in incomplete generations, with parts of the caption missing, or outputs with substantial audible artifacts. To address this, we filter out prompts that contain more than two distinct audio elements.

Table 4: Examples of prompt generation. The input caption is taken from an existing captioning dataset, while the rest is generated using an LLM. The reasoning step is not shown for brevity.

| Input Caption | Instruction | Output Caption | Elem. | Neg. Input | Neg. Output |
|---|---|---|---|---|---|
| A man speaks as birds chirp | *"Remove the birds chirping"* | A man speaks | 2 | – | birds chirp |
| Thunder and rain | *"add distant wind"* | Thunder and rain with distant wind | 2 | distant wind | – |
| A car accelerates | *"Change it to a motorcycle"* | A motorcycle accelerates | 1 | motorcycle | car |
| A plane takes off | *"it should be further away"* | A plane takes off in the distance | 1 | – | – |

**Negative Prompts.**    Depending on the edit instruction, the LLM also generates optional negative captions, which contain descriptions of audio elements that should be excluded from the input and output. These negative prompts help improve sample quality during synthetic audio generation by ensuring that elements are properly removed or added. For example, given the input caption *"A man speaks as birds chirp"*, the model generates the edit instruction *"Remove the birds chirping"* (cf. Table 4). Since this instruction removes a sound, the negative input caption is empty, while the negative output caption contains *"birds chirp"*. If the edit instruction instead adds a new sound, the negative input caption would contain that new element, while the negative output caption remains empty. Both negative input and output prompts are used during synthetic sample generation with Prompt-to-Prompt. For DDPM inversion, only the negative output prompt is used, since the input audio is not generated.

# B    Manual Edit Tasks

We define twelve editing tasks in total: ADD, REPLACE, DROP, SWAP, LOOP, PITCH, SPEED, LOW_PASS, HIGH_PASS, INPAINT, SUPER_RES, and DENOISE. These tasks differ in the number of inputs they require, the constraints on those inputs, and whether they accept controllable parameters.

Table 5: The four editing tasks that accept parameters. The function len(·) returns the audio length in seconds.

| Task/Inputs | Parameter | Description and Constraints | Example Instruction |
|---|---|---|---|
| ADD/2 | Position $t$ | Mixes a target audio into a base audio.
• $\ell = \text{len(target)} \leq L = \text{len(base)}$
• $t \in \{\text{start, middle, end}\} \cup [0, \ L - \ell]$ | *"Add the sound of a barking dog to the beginning of the street ambience."* |
| LOOP/1 | Count $l$ | Repeats an audio multiple times.
• $l \in \mathbb{Z}_{>0}$ st. len(result) $\leq$ 47s | *"Repeat five times."* |
| PITCH/1 | Semitones $p$ | Shifts pitch up or down.
• $p \in [-12, 12]$ | *"Make the voice sound deeper by three notes."* |
| SPEED/1 | Factor $s$ | Changes speed without affecting pitch.
• $s \sim \text{LogUnif}\,(1/3, 3)$
• len(result) $\leq$ 47s | *"Slow this clip down by about 30 percent."* |

**Input Constraints.** While some tasks operate on multiple audio clips, others only require one. For example, ADD overlays two audio clips, while HIGH_PASS processes a single clip by removing its low-frequency region. In addition to the number of inputs, each task may also impose constraints based on the relative length of the inputs or the number of distinct audio elements as analyzed in Section 3.1. These constraints ensure that the edited audio does not exceed the 47 second limitation of Stable Audio Open and avoids producing samples that contain an excessive number of distinct audio elements. For instance, DROP is configured to only remove a single element from a composite audio.

**Parameters.** Tasks can be separated into two groups based on whether they accept controllable parameters. The tasks ADD, LOOP, PITCH, and SPEED each accept one controllable parameter as detailed in Table 5. The values of these parameters are all numerical, except for ADD, which accepts the insertion location either as a floating point timestamp or as one of the keywords "start", "middle", or "end", which are automatically converted into their corresponding timestamp values. These parameters also come with some restrictions. For example, the insertion location of the ADD tasks cannot insert the target audio before the start or beyond the length of the base audio, while the pitch shift of the PITCH tasks is restricted to a full octave in either direction. For tasks such as LOOP that change the length of a given audio clip, the number of loops is constrained to ensure that the final audio does not exceed 47 seconds. The parameter values are sampled uniformly, unless stated otherwise, within the allowed values. The REPLACE, DROP, SWAP, LOW_PASS, HIGH_PASS, INPAINT, SUPER_RES, and DENOISE tasks do not accept parameters and are outlined in Table 6. For example, HIGH_PASS simply removes the low-frequency region of a given audio.

## C  Dataset Generation

We outline the computational resources used for generating the dataset of audio editing triplets via the three approaches introduced in Section 3. While the dataset was generated using various GPUs, we report the average inference times based on a single NVIDIA A6000 GPU for consistency. For the Prompt-to-Prompt dataset, we generate seven candidate pairs to find suitable seed and CFG values and filter them using Gemini and CLAP. For each prompt pair, this step takes on average 2.2 minutes. After finding a suitable candidate, we perform the Prompt-to-Prompt sample generation using Bayesian Optimization with 10 trials, which takes approximately 3.4 minutes per audio pair. For DDPM inversion, we use 7 Bayesian Optimization trials to find the optimal inversion parameters, taking 1.8 minutes per audio pair. To generate 100k samples (50k from each method), we used a total of 6.2k GPU hours. In contrast, generating manual edits is comparatively lightweight, as no GPUs are required. The full pipeline, including audio manipulation and generating the edit instruction, takes approximately 5.1 seconds per sample on a single CPU core.

Table 6: The eight editing tasks that take no parameters. The function $\text{len}(\cdot)$ returns the audio length in seconds and $\text{elem}(\cdot)$ counts the number of audible elements as generated in Section 3.1.

| Task/Inputs | Description and Constraints | Example Instruction |
|---|---|---|
| REPLACE/3 | Swaps out one element (target) in a composite audio (base + target) with another (replace). 
 • $\text{elem}(\text{target}) = \text{elem}(\text{base}) = 1$ 
 • $\text{len}(\text{target}), \text{len}(\text{replace}) \leq \text{len}(\text{base})$ | *"Replace the engine hum with the sound of a propeller plane."* |
| DROP/2 | Drops a sound (target) from a composite audio (base + target). 
 • $\text{elem}(\text{target}) = 1$ 
 • $\text{len}(\text{target}) \leq \text{len}(\text{base})$ | *"Remove the rain sounds from this outdoor recording."* |
| SWAP/2 | Reorders two audio clips. 
 • $\text{elem}(\text{first}) = \text{elem}(\text{second}) = 1$ 
 • $\text{len}(\text{first}) + \text{len}(\text{second}) \leq 47\text{s}$ | *"Swap the order of these two sounds."* |
| LOW_PASS/1 | Removes high-frequency region. 
 • cutoff at 8000 Hz | *"Filter out the high-pitched noise from the recording."* |
| HIGH_PASS/1 | Removes low-frequency region. 
 • cutoff at 1000 Hz | *"Remove the bass rumble from the audio."* |
| INPAINT/1 | Fills in a silent region. 
 • $\alpha \sim \mathcal{U}(0, 95)$ 
 • randomly blank out $\alpha\%$ of input | *"Restore the missing audio in the middle of this clip."* |
| SUPER_RES/1 | Reconstruct high-frequency region. 
 • input is resampled to $1/4$th of its sample rate | *"Enhance the quality of this low-frequency audio."* |
| DENOISE/1 | Removes noise from the signal. 
 • gaussian noise $\mathcal{N}(0, 0.01)$ is added to input | *"Remove the background hiss from this audio."* |

Table 7: Performance of Stable Audio Open (SAO) and its fine-tuned variant on AudioCaps. The AudioCaps test subset was used for evaluation.

| Models | FD $\downarrow$ | KL $\downarrow$ | IS $\uparrow$ | CLAP $\uparrow$ |
|---|---|---|---|---|
| SAO | 41.64 | 2.19 | 8.56±0.47 | 0.26±0.14 |
| SAO + AudioCaps | **20.85** | **1.42** | **10.05±0.51** | **0.46±0.11** |

# D   Fine-tuning

To fine-tune Stable Audio Open on both AudioCaps and our generated audio editing triplets, we follow the guidelines in the official repository [3] and adopt the default optimizer and inverse learning rate scheduler with exponential warmup. Specifically, the AdamW [31] optimizer is configured with a learning rate of $5e - 5$, $(\beta_1, \beta_2) = (0.9, 0.999)$, and weight decay of $1e - 3$. In both cases, we keep the autoencoder frozen and only train the diffusion transformer.

**Stable Audio Open on AudioCaps.** To improve prompt adherence of Stable Audio Open, we fine-tune the model on both the training and validation splits of AudioCaps [22] using a total of 47k samples. The model is trained for 15 epochs with a batch size of 64 on two NVIDIA A100 GPUs for 30 hours. For evaluation, we provide the models with captions from the test subset of AudioCaps and generate audio using 100 denoising steps and a CFG of 6. As shown Table 7, the fine-tuned version followed prompts more closely.

**SAO-Instruct.** We train the diffusion transformer on two NVIDIA A6000 GPUs with a batch size of 16 for 4 epochs. Models trained on the individual datasets (50k samples each) were trained for 30 hours, while the final model on the large combined dataset (150k samples) was trained for 80 hours.

---

[3]`https://github.com/Stability-AI/stable-audio-tools`

Table 8: Comparing two inference configurations of SAO-Instruct. Starting from an encoded input audio with added Gaussian noise preserves more input audio characteristics while still providing enough editing flexibility.

| Initial Latent | Original Ref. | | | Regenerated Ref. | | | IS ↑ | CLAP ↑ |
| | FD ↓ | LSD ↓ | KL ↓ | FD ↓ | LSD ↓ | KL ↓ | | |
| --- | --- | --- | --- | --- | --- | --- | --- | --- |
| Pure Noise | 29.73 | 2.23 | 2.59 | 19.17 | **2.61** | 2.27 | **8.10±0.69** | 0.32±0.15 |
| Audio + Noise | **18.38** | **1.36** | **0.93** | 18.97 | 2.72 | **1.76** | 7.59±1.00 | **0.38±0.14** |

Table 9: Comparison with zero-shot audio editing baselines on Production Quality (PQ) and Production Complexity (PC) from AudioBox Aesthetics [41].

| Model | PQ ↑ | PC ↑ |
| --- | --- | --- |
| AudioEditor | 5.38±0.91 | 3.02±0.72 |
| ZETA$_{T_{start}=50}$ | **6.06±0.94** | 2.76±0.69 |
| ZETA$_{T_{start}=75}$ | 6.04±0.91 | 2.73±0.66 |
| SAO-Instruct | 5.61±0.89 | **3.23±0.78** |

# E   Inference

We compare two inference configurations for SAO-Instruct in Table 8: sampling from pure noise and sampling from the Gaussian-noised latent of the encoded input audio. Starting from the noised latent substantially improves metrics computed against the original audio clips, which indicates better preservation of the input audio's characteristics. It also achieves a higher CLAP score and lower FD/KL relative to the target caption-conditioned regenerated reference, showing accurate instruction following. Overall, sampling from the noised latent preserves more input audio characteristics while still providing enough flexibility to perform the required edits.

# F   Listening Study

The evaluation interface for the subjective listening study is shown in Fig. 5. The 13 participants were volunteers and received no compensation. They were given the following instructions to rate the model outputs:

- **Quality:** How good is the sound quality of the edited audio compared to the input?
  *1 = Poor quality with strong artifacts, 5 = Same quality as the input audio*

- **Relevance:** How well does the edited audio match the given instruction?
  *1 = Completely irrelevant to instruction, 5 = Perfectly follows instruction*

- **Faithfulness:** How similar does the edited audio sound to the input audio?
  *1 = Completely different from the input audio, 5 = Same as input audio*

# G   Results

## G.1   Comparison with Baselines

To compare our model with zero-shot editing baselines, we also evaluate on objective metrics designed for automatic quality assessment of audio. Production Quality (PQ) measures technical aspects of audio quality, while Production Complexity (PC) focuses on the complexity of the audio scene [41]. The results are shown in Table 9. We observe that SAO-Instruct slightly underperforms ZETA in production quality, while slightly outperforming ZETA in production complexity. These results reflect the findings from our subjective listening study.

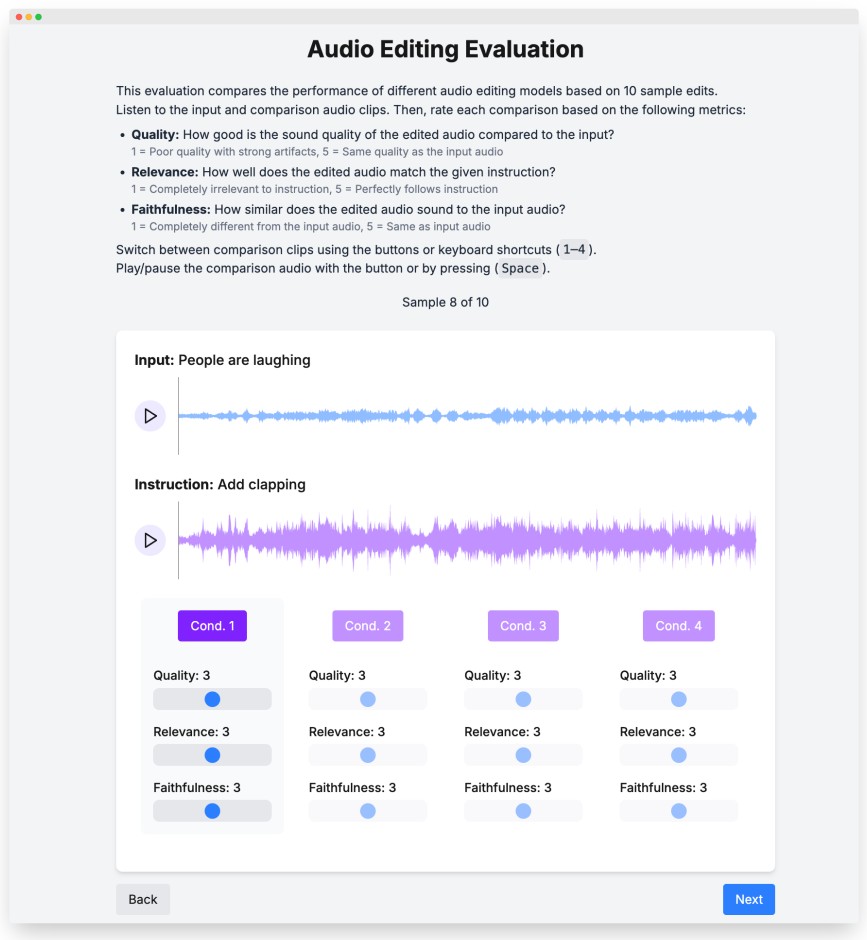

Figure 5: Evaluation interface for the subjective listening study comparing SAO-Instruct with audio editing baselines.

## G.2 Deterministic Editing

We compare SAO-Instruct with the baselines on deterministic manual editing tasks using 100 samples per edit type drawn from the AudioCaps test set. As evaluation metrics, we provide the STFT loss, the Multi-Resolution STFT loss (MR-STFT) [47], the Multi-Resolution Mel-Spectrogram loss (MR-MEL) [25], the Scale-Invariant Signal-to-Distortion Ratio (SI-SDR) [27], and the Scale-invariant Signal-to-Noise Ratio (SI-SNR) [27, 32]. Together, these metrics provide a perceptual- and signal-level view of the performance on these tasks. The input caption, edit instruction, and target captions were generated using an LLM based on the selected edit task, its optional parameter, and the captions from the selected input audio clips from AudioCaps. The edit types ADD and REPLACE are not evaluated as their target audio is ambiguous and not deterministic. For all other edit types the input audio and target audio can be created similarly as during dataset generation (cf. Section 3.4) and evaluated using common time- and frequency-based metrics [7, 40, 25]. A summary of the results averaged over all deterministic manual editing tasks is given in Table 10, while the full per-task results are shown in Table 11.

## G.3 Qualitative Results

Fig. 6 demonstrates the editing capabilities of SAO-Instruct across a range of instruction types. In the first example, ambient noise is introduced in a speech recording without distorting the primary signal. The second example shows a global transformation where the volume of rain is reduced. The

Table 10: Average performance across all deterministic manual editing tasks.

| Model | STFT ↓ | MR-STFT ↓ | MR-MEL ↓ | SI-SDR ↑ | SI-SNR ↑ |
|---|---|---|---|---|---|
| AudioEditor | 6.06±4.16 | 6.02±4.20 | 7.53±2.54 | -54.55±11.74 | -54.50±11.57 |
| ZETA$_{T_{\text{start}}=50}$ | 4.73±3.43 | 4.65±3.53 | 5.61±2.12 | -28.75±13.68 | -28.89±13.65 |
| ZETA$_{T_{\text{start}}=75}$ | 5.21±4.41 | 5.13±4.51 | 5.91±2.19 | -30.65±14.85 | -30.67±14.83 |
| SAO-Instruct | **2.91±1.82** | **2.82±1.83** | **3.47±1.33** | **-21.62±12.81** | **-21.79±12.65** |

third example shows a more localized operation, in which the sound of crumpling paper is replaced with typing, while the original speech remains intact. Notably, the model performs all edits based solely on the input audio and a free-form edit instruction, with no access to either the original or target audio captions.

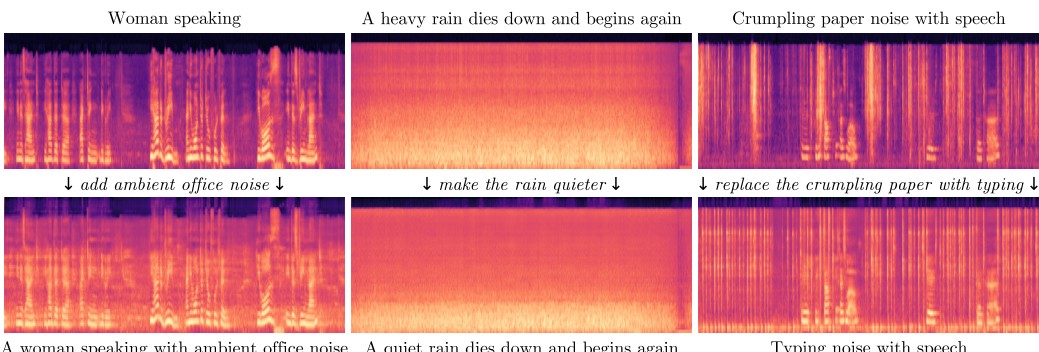

Figure 6: Edits performed by SAO-Instruct. The model has only access to the input audio and the edit instruction. It is able to perform global operations and local operations while keeping the overall background context intact.

**Failure Cases**   While the performance of SAO-Instruct can be further improved by per-sample adjustments, such as tuning the CFG scale or the amount of noise applied to the initial encoded audio, some limitations remain. In Fig. 7, we observe that the phrasing of edit instruction can influence the edit quality and accuracy of the model. The model also occasionally struggles to reconstruct coherent speech and may produce edits with audible artifacts. In Fig. 8, when adding elements, the newly added sounds sometimes fail to naturally blend in with the background and instead appear overlaid on existing sound elements. Additionally, if a clip contains many distinct elements, the model may be unable to alter sounds or confuses them, leading to unintended edits. These limitations primarily stem from insufficient data diversity and could be mitigated by training on larger and more diverse datasets.

# H   Broader Impacts

Our work introduces SAO-Instruct, a model that enables free-form instruction-based audio editing. It enables an intuitive and accessible way for users to edit audio using natural language instructions. However, it also introduces potential risks of misuse, such as the manipulation of audio clips for deceptive or harmful purposes. To mitigate these concerns, we do not scrape data from arbitrary sources and only use well-established datasets that have been widely adopted in prior research. We encourage future work to further explore responsible deployment practices for instruction-based audio models and methods for detecting synthetically edited audio.

Table 11: Performance on all deterministic manual editing tasks.

| Task | Model | STFT ↓ | MR-STFT ↓ | MR-MEL ↓ | SI-SDR ↑ | SI-SNR ↑ |
|---|---|---|---|---|---|---|
| DROP | AudioEditor | 6.59±6.63 | 6.58±6.69 | 6.37±2.67 | -51.24±9.26 | -51.20±9.28 |
| | ZETA$_{T_{\text{start}}=50}$ | **5.58±7.22** | **5.57±7.63** | 5.51±2.17 | -20.21±16.19 | -20.38±16.56 |
| | ZETA$_{T_{\text{start}}=75}$ | 6.27±8.88 | 6.29±9.41 | 5.93±2.31 | -23.28±18.21 | -23.06±17.93 |
| | SAO-Instruct | 6.12±11.63 | 6.06±11.68 | **4.44±3.08** | **-13.77±16.86** | **-13.69±16.75** |
| SWAP | AudioEditor | 6.22±1.13 | 6.07±1.11 | 11.78±2.71 | -60.70±13.89 | -59.94±13.77 |
| | ZETA$_{T_{\text{start}}=50}$ | 4.39±1.34 | 4.29±1.31 | 6.70±2.30 | -57.52±10.98 | -58.01±10.28 |
| | ZETA$_{T_{\text{start}}=75}$ | 4.56±1.58 | 4.46±1.56 | **6.45±1.72** | -56.45±9.86 | -56.53±9.65 |
| | SAO-Instruct | **4.38±1.05** | **4.25±1.04** | 7.32±2.17 | **-54.19±11.35** | **-54.30±11.36** |
| LOOP | AudioEditor | 6.34±2.23 | 6.21±2.19 | 10.39±3.90 | -55.31±12.19 | -55.19±11.99 |
| | ZETA$_{T_{\text{start}}=50}$ | 4.70±1.85 | 4.55±1.83 | 8.01±3.41 | -18.13±12.79 | -18.10±12.75 |
| | ZETA$_{T_{\text{start}}=75}$ | 5.11±2.29 | 4.95±2.27 | 8.26±3.34 | -19.61±14.10 | -19.58±14.28 |
| | SAO-Instruct | **2.01±0.58** | **1.94±0.58** | **2.36±0.93** | **-11.91±13.78** | **-11.89±13.76** |
| PITCH | AudioEditor | 5.73±4.04 | 5.73±4.20 | 5.79±1.97 | -51.73±11.31 | -51.40±10.97 |
| | ZETA$_{T_{\text{start}}=50}$ | 5.44±7.75 | 5.42±8.32 | 4.78±1.55 | -46.78±13.71 | -46.79±13.79 |
| | ZETA$_{T_{\text{start}}=75}$ | 5.95±8.82 | 5.94±9.44 | 5.08±1.85 | -48.66±14.11 | -48.59±14.21 |
| | SAO-Instruct | **2.63±0.70** | **2.49±0.71** | **3.11±0.96** | **-41.39±14.74** | **-41.31±14.42** |
| SPEED | AudioEditor | 6.49±4.85 | 6.45±4.98 | 7.02±3.03 | -50.96±13.55 | -50.60±12.52 |
| | ZETA$_{T_{\text{start}}=50}$ | 4.94±2.82 | 4.83±2.83 | 5.84±2.44 | -45.93±13.81 | -46.75±13.66 |
| | ZETA$_{T_{\text{start}}=75}$ | 5.33±2.97 | 5.23±2.99 | 6.22±2.54 | -47.28±12.97 | -47.39±12.85 |
| | SAO-Instruct | **3.53±1.23** | **3.43±1.26** | **5.13±1.37** | **-45.12±13.79** | **-46.15±13.00** |
| LOW_PASS | AudioEditor | 4.94±2.17 | 4.92±2.16 | 5.92±1.90 | -53.40±10.43 | -53.99±10.64 |
| | ZETA$_{T_{\text{start}}=50}$ | 3.41±1.27 | 3.32±1.27 | 4.87±1.76 | -16.07±13.30 | -15.99±13.30 |
| | ZETA$_{T_{\text{start}}=75}$ | 3.79±1.54 | 3.70±1.48 | 5.28±1.95 | -17.76±15.49 | -17.78±15.47 |
| | SAO-Instruct | **1.63±0.45** | **1.56±0.47** | **1.89±0.59** | **-2.26±10.95** | **-2.28±10.95** |
| HIGH_PASS | AudioEditor | 6.95±7.75 | 6.94±7.74 | 6.52±1.46 | -62.28±12.08 | -62.25±12.08 |
| | ZETA$_{T_{\text{start}}=50}$ | 4.89±3.42 | 4.82±3.43 | 5.15±1.77 | -31.05±12.92 | -30.98±13.01 |
| | ZETA$_{T_{\text{start}}=75}$ | 5.82±5.53 | 5.71±5.46 | 5.65±2.12 | -34.64±15.85 | -34.23±15.52 |
| | SAO-Instruct | **1.75±0.38** | **1.68±0.40** | **2.08±0.59** | **-28.48±12.66** | **-28.48±12.66** |
| INPAINT | AudioEditor | 6.50±3.80 | 6.43±3.91 | 9.81±4.40 | -54.40±12.39 | -54.71±11.92 |
| | ZETA$_{T_{\text{start}}=50}$ | 4.61±3.45 | 4.54±3.44 | 5.33±2.13 | -22.34±17.05 | -22.51±17.09 |
| | ZETA$_{T_{\text{start}}=75}$ | 5.62±6.32 | 5.53±6.28 | 5.13±1.74 | -23.49±16.82 | -24.21±17.06 |
| | SAO-Instruct | **2.49±1.00** | **2.42±0.99** | **3.31±2.24** | **-13.98±13.20** | **-14.61±12.85** |
| SUPER_RES | AudioEditor | 5.56±2.66 | 5.53±2.66 | 6.66±1.75 | -52.83±12.23 | -52.93±12.31 |
| | ZETA$_{T_{\text{start}}=50}$ | 4.51±3.02 | 4.43±3.02 | 4.68±1.63 | -14.19±11.74 | -14.18±11.75 |
| | ZETA$_{T_{\text{start}}=75}$ | 4.75±3.47 | 4.67±3.47 | 4.96±1.70 | -16.97±14.71 | -16.96±14.74 |
| | SAO-Instruct | **2.16±0.71** | **2.10±0.71** | **2.36±0.81** | **-1.68±9.70** | **-1.67±9.69** |
| DENOISE | AudioEditor | 5.30±6.38 | 5.29±6.38 | 5.00±1.66 | -52.65±10.11 | -52.79±10.24 |
| | ZETA$_{T_{\text{start}}=50}$ | 4.79±2.17 | 4.70±2.17 | 5.28±2.02 | -15.31±14.27 | -15.16±14.28 |
| | ZETA$_{T_{\text{start}}=75}$ | 4.91±2.70 | 4.83±2.70 | 6.10±2.59 | -18.37±16.37 | -18.34±16.62 |
| | SAO-Instruct | **2.36±0.50** | **2.30±0.48** | **2.74±0.56** | **-3.39±11.03** | **-3.50±11.02** |

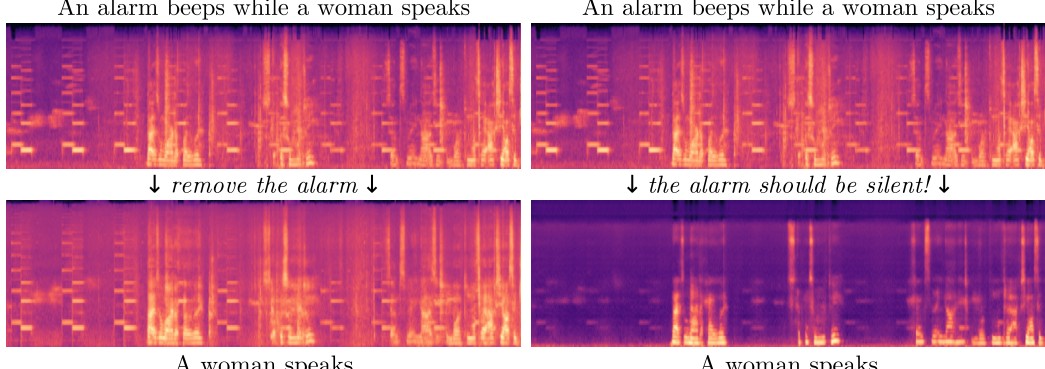

Figure 7: An example failure case where the phrasing of an instruction impacts edit quality and accuracy. While *"remove the alarm"* fails to suppress the alarm, *"the alarm should be silent!"* is more successful.

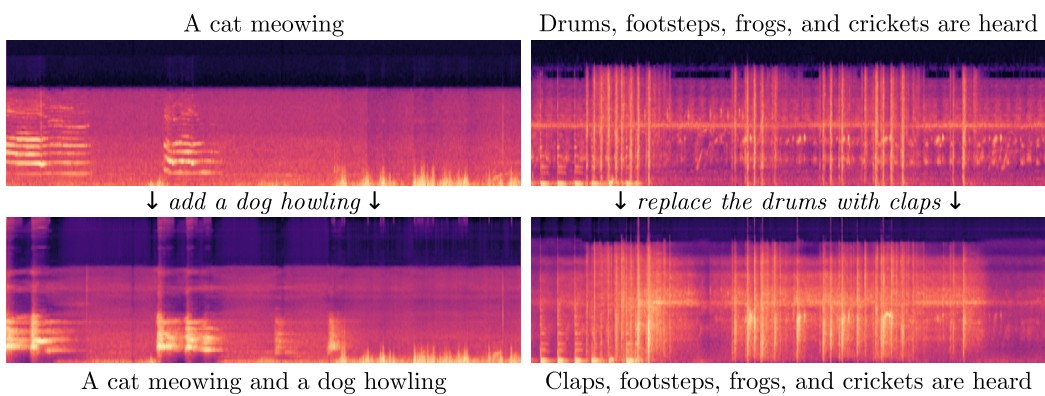

Figure 8: Examples of failure cases where newly added sounds fail to blend naturally with the background and where edits in complex audio scenes lead to unintended edits.

## I  Licenses

We provide the licenses of datasets and models that we build upon in our work below. We ensure that our use of these assets fully complies with their license and terms of use. AudioCaps [22] is released under the MIT License. AudioSet [12] and AudioSet-SL [15] are available under the CC BY 4.0 license. WavCaps [35] is available for academic use and includes audio data from multiple sources. We follow the licensing terms for the BBC Sound Effects [4] library and respect the licenses associated with each audio clip from FreeSound.[5] We comply with the Stability AI Community License Agreement [6], which allows the use of Stable Audio Open [9] for research purposes. CLAP [45] is licensed under the Creative Commons CC0 1.0 Universal license, which allows unrestricted use and distribution.

---

[4] https://sound-effects.bbcrewind.co.uk/licensing
[5] https://freesound.org/help/faq/#licenses
[6] https://huggingface.co/stabilityai/stable-audio-open-1.0

