# OpenReview forum: "SAO-Instruct: Free-form Audio Editing using Natural Language Instructions"
_NeurIPS.cc/2025/Conference — NeurIPS 2025 poster_

### Official Review · Reviewer_mRyx · 2025-06-15

**Clarity:** 4
**Significance:** 4
**Originality:** 4
**Rating:** 6
**Confidence:** 4

**Summary:**

This paper introduces SAO-Instruct, an audio input + text editing instruction to edited audio output framework. The model is finedtuning the Stable Audio Open (a latent diffusion model model released by Stability AI in July 2024) on a mixture composed of partial synthetic data, fully synthetic data, and real data (manual edits). The performance and latency beats the AudioEditor and ZETA baselines on multiple metrics. Compared to other methods, the proposed approach is particularly great at "better preserving the characteristics and quality of the input audio".

**Questions:**

Question 1: CLAP score is used in many places in the paper, including training data generation, Objective Function, and evaluations. Is there a risk of overfitting a specific CLAP model here? Any mitigations?

Question 2: Manual edit - is this really manualy edit or agentic edit? It's not super clear to me how manual edit was implemented. It seems each manual edit is just from a list of predefined function calls with arguments. But how are the arguments determined? Using LLM or by human?

Question 3: Does the authors plan to open source the generated data?

Question 4: In Table 3, are all the models being compared here trained using completely different data? Clarification would be helpful.

**Ethical Concerns:**

["NO or VERY MINOR ethics concerns only"]

**Final Justification:**

My questions have all been resolved. Really strong paper, with huge potentials for real world applications. My final rating is 6 Strong Accept.

**Limitations:**

A study of failure cases would be helpful. Also in the demo site, adding a section for failure cases or cases where the proposed model performs worse than baseline models will complete the story.

A discussion regarding the "safety" aspects of the proposed model might be helpful. E.g. what can we do to prevent evil or inappropriate use of the model?

**Paper Formatting Concerns:**

No concern.

**Quality:**

4

**Strengths And Weaknesses:**

Strengths:
The paper is very clearly presented and is very easy to follow. I find two parts of the paper particularly interesting and inspiring to the audience: the data generation part and the sample selection/rejection part. The paper used SAO to generate two types of data: partially synthetic data (input audio is real, output audio is SAO) and fully synthetic data (both input and output audio are SAO). For sample selection, authored first use Gemini to perform perceptual quality check on the data, then used CLAP score to select.

The demos, especially the spectrogram view, are super convincing. Well done!

Weaknesses:

1 - Several metrics were not on-par with the AudioEditor and ZETA baselines. But given the much lower latency, I do not really see this as a big weakness.

2 - Also, all the baselines compared in the paper are diffusion based model. No autoregressive models were compared.

3 - The test sets are generated in the same way as the training sets for the proposed method, which creates a potentially unfair comparison with the baseline models.

---

> ### Author Rebuttal · Authors · 2025-07-30
>
> We thank the reviewer for their valuable insights and address their feedback in the following:
>
> > Question 1: CLAP score is used in many places in the paper, including training data generation, Objective Function, and evaluations. Is there a risk of overfitting a specific CLAP model here? Any mitigations?
>
> We acknowledge the risk of over‑reliance on a single CLAP model. In our pipeline, CLAP is one metric among several: Prompt-to-Prompt candidate search also uses a perceptual Gemini filter, the Bayesian Optimization objective includes a multi-scale mel-spectrogram distance, and evaluation includes reference-based metrics in addition to a human listening study. In our experiments we used different public CLAP models and checkpoints (e.g., LAION and Microsoft), and while absolute scores were slightly different, the produced samples and rankings were overall stable. While we did not observe overfitting to the chosen CLAP checkpoint, future work could explore aggregating scores across diverse CLAP models and checkpoints to further reduce model-specific bias.
>
> > Question 2: Manual edit - is this really manualy edit or agentic edit? It's not super clear to me how manual edit was implemented. It seems each manual edit is just from a list of predefined function calls with arguments. But how are the arguments determined? Using LLM or by human?
>
> Yes, edits are randomly sampled from a predefined set of supported edit operations (e.g., ADD, PITCH, ...). Some of these editing tasks support additional parameters as outlined in Table 5. For example, the PITCH tasks allows to change the amount of pitch shift. Unless stated otherwise, parameter values are sampled uniformly from a fixed range of supported values and the edit is deterministically applied to the audio. The LLM does not choose the operation or its arguments but is responsible for creating a fitting edit instruction for the sampled editing task and its optional parameter. This is done in multiple stages: first, the initial edit instruction is generated, and then optionally rephrased and minimized to make it more natural. In the revised manuscript, we clarified the role of the LLM to avoid confusion.
>
> > Question 3: Does the authors plan to open source the generated data?
>
> We will open source the model and our full codebase. We are also exploring releasing the generated datasets. For the semi-synthetic DDPM inversion and the fully real manual edits we will first need to verify potential licensing issues as they contain audio from FreeSound and AudioSet-SL.
>
> > Question 4: In Table 3, are all the models being compared here trained using completely different data? Clarification would be helpful.
>
> We thank the reviewer for raising this point. AudioEditor uses Auffusion [1] as its underlying model, fine-tuned on several general audio datasets including AudioCaps, WavCaps, ESC50, and MACS. The ZETA DDPM inversion approach uses Stable Audio Open as the base model, trained on Freesound and FMA. We have updated Section 5.2 to make these training data differences more explicit.
>
> > A study of failure cases would be helpful. Also in the demo site, adding a section for failure cases or cases where the proposed model performs worse than baseline models will complete the story.
>
> We thank the reviewer for this suggestion. We will update the demo page with relevant failure cases after the review period ends (NeurIPS rebuttal policy).
>
> > A discussion regarding the "safety" aspects of the proposed model might be helpful. E.g. what can we do to prevent evil or inappropriate use of the model?
>
> We discuss broader impacts in Appendix G, including that we do not scrape data from arbitrary sources and rely on well-established datasets which have been widely adopted in prior research. According with the reviewer’s feedback, our revised manuscript will expand on this section to discuss possible safeguards against inappropriate use of the model. For example, embedding inaudible watermarks in the model’s output to help detect AI-generated content, similar to existing watermarking approaches [2, 3, 4]. Additionally, for our current model checkpoints, we will consider a more restricted release of the model weights under a research license, to further reduce the risk of misuse.
>
>
> [1] Xue, Jinlong, et al. "Auffusion: Leveraging the power of diffusion and large language models for text-to-audio generation." IEEE/ACM Transactions on Audio, Speech, and Language Processing (2024).
>
> [2] San Roman, Robin, et al. "Latent watermarking of audio generative models." ICASSP 2025-2025 IEEE International Conference on Acoustics, Speech and Signal Processing (ICASSP). IEEE, 2025.
>
> [3] Wen, Yuxin, et al. "Tree-ring watermarks: Fingerprints for diffusion images that are invisible and robust." arXiv preprint arXiv:2305.20030 (2023).
>
> [4] Guo, Yiyang, et al. "FreqMark: Invisible Image Watermarking via Frequency Based Optimization in Latent Space." Advances in Neural Information Processing Systems 37 (2024): 112237-112261.

---

> > ### Comment · Reviewer_mRyx · 2025-08-01
> >
> > Thanks for the responses.
> >
> > > Unless stated otherwise, parameter values are sampled uniformly from a fixed range of supported values and the edit is deterministically applied to the audio. The LLM does not choose the operation or its arguments but is responsible for creating a fitting edit instruction for the sampled editing task and its optional parameter.
> >
> > I am really confused by this. You are using manual edits to prepare / create samples that are later used for training. That means, for a specific LLM-generated edit instruction, the target edited audio (as training target) is created by applying manual edits. My question is, how do you determine for this specific edit instruction, what manually edit operations shall be used, and each with what parameters? How can the parameters be uniformly sampled (as said in your rebuttal)?
> >
> > Or, are you actually saying, you first sample random parameters of the manual edit options, then use LLM to create the edit instruction based on these parameters? If that is the case, how do you determine the number and order of manual edits to use? Also, if that is the case, Figure 2 is misleading.
> >
> > I would suggest following changes to the revised manuscript:
> > 1. Clearly point out that the limitations of each evaluation / metric you used in the paper. E.g. CLAP score -> same CLAP model used in training.
> > 2. Open source the generated datasets (or a subset where license allows).
> > 3. Clarify the different training data used for each baseline model.
> > 4. Add case studies for failure cases.

---

> > > ### Author Response · Authors · 2025-08-02
> > >
> > > We thank the reviewer for the follow-up and appreciate the chance to clarify the manual edit pipeline, which is one of three distinct data generation pipelines.
> > >
> > > When generating a triplet (input audio, edit instruction, output audio) from the manual edit pipeline, we begin by sampling one of the 12 edit operations with equal probability. Suppose the sampled task is PITCH, which works on a single audio clip and has one numeric argument: the semitone shift $p \in [-12, 12]$. Because PITCH takes exactly one clip, we then draw a random example from FreeSound, for instance a recording with the caption “birds chirping”. Next, we uniformly sample $p$ from the allowed range. In this example, let’s assume we get a value of $p=3$.
> > >
> > > Up to this point no language model has been involved. We now give this information (operation = PITCH, $p = 3$, caption = “birds chirping”) to the LLM, and instruct it to generate an appropriate edit instruction. This could result in “Make the birds chirping sound three semitones higher”. To increase diversity, we sometimes request a rephrase and a condensed version of that instruction. After these steps, we apply the deterministic edit (operation = PITCH, $p = 3$) to the audio by shifting the clip up 3 semitones, which yields the target edited audio.
> > >
> > > Finally, the triplet consisting of the original clip, the generated instruction, and the edited clip is added to the training set. We hope this resolves any confusion. If not, we will gladly provide further clarification.
> > >
> > > > Also, if that is the case, Figure 2 is misleading.
> > >
> > > We have updated the figure in the revised manuscript to clarify that we first sample an edit operation before the input audio clips are sampled. This is due to input constraints of certain operations (e.g., PITCH takes one audio clip, REPLACE requires three audio clips)
> > >
> > > > I would suggest following changes to the revised manuscript
> > >
> > > We appreciate these suggestions. We have incorporated and addressed these points in the revised manuscript.

---

### Official Review · Reviewer_AzCo · 2025-06-17

**Clarity:** 2
**Significance:** 2
**Originality:** 3
**Rating:** 4
**Confidence:** 5

**Summary:**

SAO-Instruct is an instruction-based audio editing framework that fine-tunes Stable Audio Open’s diffusion transformer on a dataset of editing triplets generated via Prompt-to-Prompt, DDPM inversion, and manual DSP edits, enabling users to modify existing clips with plain-English instructions.

**Questions:**

See weaknesses mentioned above.

**Ethical Concerns:**

["NO or VERY MINOR ethics concerns only"]

**Final Justification:**

I will upgrade my score from 2 to 4 as the authors provided separate evaluation results for 10 types of editing operations.

**Limitations:**

In the limitations sections, the authors mentioned their approach “can naturally extend to music”. The training data is drawn from event‐focused caption datasets (AudioCaps, WavCaps, AudioSet) and manually edited effects. None of these guarantee coverage of the complex patterns found in music. For example, you are not able to simply construct a regenerated reference of the instruction “add a guitar track to this song”. The regenerated reference pipeline ignores the source audio. It doesn’t know the melody, key, or ambience of your particular song, so it will produce a new guitar-backed track rather than a faithful edit of your input.
Thus, I am strongly against to the statement that their approach “can naturally extend to music”.

**Paper Formatting Concerns:**

No Paper Formatting Concerns.

**Quality:**

2

**Strengths And Weaknesses:**

Strengths:
- Free-form, instruction-based editing.
- Combines three complementary data-generation pipelines, fully synthetic Prompt-to-Prompt, DDPM inversion, and real manual DSP edits.
- Fast inference. At ≈9.94 s per 10 s clip on an NVIDIA A6000 GPU, SAO-Instruct is nearly 8× faster than AudioEditor.

Weaknesses:
- The “regenerated reference” is produced solely by sampling from Stable Audio Open using the target caption, with no conditioning on the original input clip. This isn’t a true “ground-truth” edit of the input audio, but rather a brand-new, synthetic sample that may lose all source-specific content.
- In Section 4.2 the authors define FD/LSD/KL (and IS, CLAP) as their objective metrics, but these are always computed and reported as a single number averaged over all instructions. I want to point out that using the same set of evaluation criteria for different editing operations is not reasonable. For operations like LOOP with instructions as “repeat the audio 5 times”, we want the model to repeat the exact same audio 5 times. I would expect the edited audio to be exactly the same as the ground truth to be considered as a successful edit. However, none of the metrics in Table 2 and 3 are able to provide information like this because the performance values are averaged over different operations. While LOOP is defined as “repeat the audio l times” in Table 5 (with the constraint len(result) ≤ 47 s), there is no mechanism in the evaluation to verify how many loops were actually produced. A perfect DSP-based loop would trivially get an exact-match score, but the paper’s FD/LSD/KL cannot distinguish “five perfect repeats” from “four repeats plus a cut-off,” nor do they introduce any operation-specific metric (e.g. loop-count accuracy). I don’t think simply relying on subjective evaluation for these kinds of requirements are suitable as they are not precise enough.
- The subjective evaluation is unreliable due to these two reasons:
1. The scores are based on a 1 to 5 rating, while only 1 and 5 are provided with a criteria. 2, 3 and 4 are undefined. Human annotators do not have a clear criteria for the ratings. Please refer to AudioBox on how to define criteria for ratings of 1 to 5.
2. The design of Faithfulness in Section 4.2 (Subjective Metrics) is not reasonable and very confusing. Faithfulness is defined as “the similarity between the reference and the edited audio” in Section 4.2, while mentioned in Appendix, it is defined as “How similar does the edited audio sound to the input audio?” By simply reading through these statements, one would consider the term “reference” referring to the ground truth edited audio, while “input audio” is the original audio intended to be edited. Even if there is no confusion in the term “reference”, setting the criteria of faithfulness as “How similar does the edited audio sound to the input audio?” makes no sense because isn’t the goal of audio editing intended to modify the input audio into something different? Why do we need a rating of 5 indicating that the edited audio is “same as input audio”, which means that a rating of 5 indicates that the model did not do edits, while this metric is the higher the better?
- Instructions are generated via a templated LLM pipeline and evaluated only in English. The paper does not explore robustness to paraphrases beyond its few-shot LLM variations, nor to multi-step or compound edits (for example “speed up then repeat”).

---

> ### Author Rebuttal · Authors · 2025-07-30
>
> We thank the reviewer for their valuable insights and address their feedback in the following:
>
> > The “regenerated reference” is produced solely by sampling from Stable Audio Open using the target caption, with no conditioning on the original input clip. This isn’t a true “ground-truth” edit of the input audio, but rather a brand-new, synthetic sample that may lose all source-specific content.
>
> We agree that a regenerated reference based on the target caption is not a ground-truth edit. Evaluating the edit accuracy is challenging as paired ground-truth edited general audio datasets are unavailable. This is why we report two complementary proxies in accordance with previous work [1]: (i) scores against the original audio clips to measure quality and content preservation to guard against over-editing, and (ii) scores against this regenerated reference conditioned on the target caption to approximate edit accuracy. These metrics should be interpreted jointly to assess the model’s ability to perform accurate and faithful edits. To compare our model with other baselines, we additionally perform a human listening study to reduce reliance on these reference-based metrics.
>
> > In Section 4.2 the authors define FD/LSD/KL (and IS, CLAP) as their objective metrics, but these are always computed and reported as a single number averaged over all instructions. I want to point out that using the same set of evaluation criteria for different editing operations is not reasonable. For operations like LOOP with instructions as “repeat the audio 5 times”, we want the model to repeat the exact same audio 5 times. I would expect the edited audio to be exactly the same as the ground truth to be considered as a successful edit. However, none of the metrics in Table 2 and 3 are able to provide information like this because the performance values are averaged over different operations. While LOOP is defined as “repeat the audio l times” in Table 5 (with the constraint len(result) ≤ 47 s), there is no mechanism in the evaluation to verify how many loops were actually produced. A perfect DSP-based loop would trivially get an exact-match score, but the paper’s FD/LSD/KL cannot distinguish “five perfect repeats” from “four repeats plus a cut-off,” nor do they introduce any operation-specific metric (e.g. loop-count accuracy). I don’t think simply relying on subjective evaluation for these kinds of requirements are suitable as they are not precise enough.
>
> We agree that for deterministic edits such as LOOP operation-specific evaluation measures (e.g., loop-count accuracy) are more appropriate then our chosen reference-based metrics. Our work focuses on free-form instruction-based editing, where edits vary widely in type of strength and are therefore difficult to categorize and evaluate. This motivated our choice of using reference proxies to that measure (i) similarity to original audio clips to measure quality and content preservations, and (ii) similarity to a regenerated sample conditioned on the target caption to approximate instruction relevance. To evaluate aspects not captured by these reference proxies, we additionally conducted a subjective listening study. We will clarify in the revised manuscript that aggregate metrics are intended to be interpreted jointly and will note the value of task-specific validators (e.g., loop-count checks) for future benchmarks.
>
> > The scores are based on a 1 to 5 rating, while only 1 and 5 are provided with a criteria. 2, 3 and 4 are undefined. Human annotators do not have a clear criteria for the ratings. Please refer to AudioBox on how to define criteria for ratings of 1 to 5.
>
> We thank the reviewer for this suggestion. All participants in our listening study were recruited from our lab and briefed prior to the study. While they were familiar with such listening tests, we acknowledge that providing text anchors for every point on the 1-to-5 scale can improve consistency and reduce potential confusion. We will adopt this practice in future listening tests.
>
> > The design of Faithfulness in Section 4.2 (Subjective Metrics) is not reasonable and very confusing. Faithfulness is defined as “the similarity between the reference and the edited audio” in Section 4.2, while mentioned in Appendix, it is defined as “How similar does the edited audio sound to the input audio?” By simply reading through these statements, one would consider the term “reference” referring to the ground truth edited audio, while “input audio” is the original audio intended to be edited.
>
> We thank the reviewer for pointing out this ambiguity. We will revise the explanation of the subjective metrics in Section 4.2 to use the term “input audio” instead of “reference”. For clarity, we will explicitly state that “input audio” refers to the original audio provided for editing, as described in the instructions given to the listening study participants.
>
> > Even if there is no confusion in the term “reference”, setting the criteria of faithfulness as “How similar does the edited audio sound to the input audio?” makes no sense because isn’t the goal of audio editing intended to modify the input audio into something different? Why do we need a rating of 5 indicating that the edited audio is “same as input audio”, which means that a rating of 5 indicates that the model did not do edits, while this metric is the higher the better?
>
> Because evaluating Faithfulness or Relevance in isolation can be misleading, we interpret them together: Faithfulness quantifies how many acoustic characteristics of the original clip survive the edit, whereas Relevance measures how accurately the edit instruction is carried out. For example, consider the input “birds chirping with water flowing” and the instruction “add dogs barking."
> - If the output keeps the birds and water and adds dogs, both metrics are high.
> - If birds disappear but dogs are added, Relevance stays high while Faithfulness drops.
> - If dogs are never added, Faithfulness may remain high yet Relevance is low.
>
> As shown in previous work [1], jointly examining these two metrics reveals whether the model both follows the instruction and preserves the essence of the source audio.
>
> > Instructions are generated via a templated LLM pipeline and evaluated only in English. The paper does not explore robustness to paraphrases beyond its few-shot LLM variations, nor to multi-step or compound edits (for example “speed up then repeat”).
>
> We thank the reviewer for raising this point. Because our underlying model (Stable Audio Open) was mainly trained on English data, we restricted our pipeline and evaluation to English instructions. While our work focuses on single free-form edits, we agree that compound edits are an interesting direction for future work. We will note these limitations in the revised manuscript.
>
> > In the limitations sections, the authors mentioned their approach “can naturally extend to music”. The training data is drawn from event‐focused caption datasets (AudioCaps, WavCaps, AudioSet) and manually edited effects. None of these guarantee coverage of the complex patterns found in music. For example, you are not able to simply construct a regenerated reference of the instruction “add a guitar track to this song”. The regenerated reference pipeline ignores the source audio. It doesn’t know the melody, key, or ambience of your particular song, so it will produce a new guitar-backed track rather than a faithful edit of your input. Thus, I am strongly against to the statement that their approach “can naturally extend to music”.
>
> The regenerated reference is used only as an evaluation proxy to measure edit accuracy. During training, SAO-Instruct receives the input audio by concatenating it to the model’s input channels. During inference, the input audio is additionally encoded into the latent space, with added Gaussian noise, and used as a starting point for the denoising process. The amount of Gaussian noise can be configured and controls how similar the edited audio should be to the provided input audio.
>
> Regarding applicability to music, the same semi-synthetic triplet generation (input audio, edit instruction, output audio) can be applied on music datasets using a music-specific generative model. Editing music using DDPM inversion was already demonstrated in [2]. To avoid overstating the claim, we have revised our statement in the updated manuscript to: “While our work focuses on editing general audio, our approach could be extended to music editing tasks provided appropriate generative music models exist and triplets are constructed from music datasets.”
>
> Based on our feedback and proposed improvements, would the reviewer be willing to reconsider their score?
>
> [1] Jia, Yuhang, et al. "AudioEditor: A training-free diffusion-based audio editing framework." ICASSP 2025-2025 IEEE International Conference on Acoustics, Speech and Signal Processing (ICASSP). IEEE, 2025.
>
> [2] Manor, Hila, and Tomer Michaeli. "Zero-shot unsupervised and text-based audio editing using DDPM inversion." arXiv preprint arXiv:2402.10009 (2024).

---

> ### Comment · Reviewer_AzCo · 2025-08-01
>
> The reason I insist on using different sets of evaluation criteria for different editing operations is that some editing operations are easy, ex: loop. Averaging the metric results over all kinds of editing operations does not provide a clear picture of how each kind of operation performs. The results of this work are presented as an overall score across different edits. Simple editing operations can easily contribute to the overall score and make the metric values look good. In order to raise my score, I would need to see performance values for each different kind of editing operation. This cannot be simply addressed by listing the suggested items of the reviewer as future work.
>
> The authors admit, "for deterministic edits, such as LOOP operation-specific evaluation measures (e.g., loop-count accuracy) are more appropriate than our chosen reference-based metrics." In Appendix B, the authors also explicitly define twelve editing tasks in total. This does not align with their claim in the rebuttal, saying "this work focuses on single free-form edits". Even if we pivot this work to be able to generalize to free-form edits, the concerns can still be simply addressed by reporting evaluation scores for each editing task they define.
>
> Additionally, three of the concerns mentioned in the review are listed as "future work" in the rebuttal. Thus, I decide to keep my score at this stage if they do not address the main concerns.

---

> > ### Author Response · Authors · 2025-08-02
> >
> > We thank the reviewer for their comment. We appreciate the opportunity to provide additional feedback.
> >
> > As the reviewer requested, here are the results for the following experiment: We evaluated the models on all manual edit tasks (except ADD & REPLACE, more on this below), using 100 samples per edit type drawn from the AudioCaps test set. As evaluation metrics, we provide the STFT loss, the Multi-Resolution STFT (MR-STFT), the Multi-Resolution Mel-Spectrogram (MR-MEL), the SI-SDR, and the SI-SNR. These metrics provide a perceptual- and signal-level view of the model performance on these tasks. The input caption, edit instruction, and target captions were generated using an LLM based on the selected edit task, its optional parameter, and the captions from the selected input audio clips from AudioCaps (e.g., “An engine running“ -> “Repeat the engine running in a loop 3 times”).
> >
> > **Results averaged over all tasks:**
> >
> > | Model        |    STFT $\downarrow$     |   MR-STFT $\downarrow$   |   MR-MEL $\downarrow$    |      SI-SDR $\uparrow$      |      SI-SNR $\uparrow$      |
> > |:-------------|:------------------------:|:------------------------:|:------------------------:|:---------------------------:|:---------------------------:|
> > | AudioEditor  |     $6.06 \pm 4.16$      |     $6.02 \pm 4.20$      |     $7.53 \pm 2.54$      |     $-54.55 \pm 11.74$      |     $-54.50 \pm 11.57$      |
> > | ZETA_50      |     $4.73 \pm 3.43$      |     $4.65 \pm 3.53$      |     $5.61 \pm 2.12$      |     $-28.75 \pm 13.68$      |     $-28.89 \pm 13.65$      |
> > | ZETA_75      |     $5.21 \pm 4.41$      |     $5.13 \pm 4.51$      |     $5.91 \pm 2.19$      |     $-30.65 \pm 14.85$      |     $-30.67 \pm 14.83$      |
> > | SAO-Instruct | $\mathbf{2.91 \pm 1.82}$ | $\mathbf{2.82 \pm 1.83}$ | $\mathbf{3.47 \pm 1.33}$ | $\mathbf{-21.62 \pm 12.81}$ | $\mathbf{-21.79 \pm 12.65}$ |
> >
> > The edit types ADD and REPLACE are not evaluated as their target audio is ambiguous (e.g., how exactly should the dogs sound like for “add dogs barking”). For all other edit types the input audio and target audio can be created during evaluation equivalently as done during dataset generation (see Section 3.4) and therefore evaluated using commonly-used time- and frequency-based metrics [1,2,3]. The task-specific results are provided in the comments below.
> >
> > Do these results address the reviewer’s concerns?
> >
> > [1] Défossez, Alexandre, et al. "Demucs: Deep extractor for music sources with extra unlabeled data remixed." arXiv preprint arXiv:1909.01174 (2019).
> >
> > [2] Shin, Ui-Hyeop, et al. "Separate and reconstruct: Asymmetric encoder-decoder for speech separation." Advances in Neural Information Processing Systems 37 (2024): 52215-52240.
> >
> > [3] Kumar, Rithesh, et al. "High-fidelity audio compression with improved RVQGAN." Advances in Neural Information Processing Systems 36 (2023): 27980-27993.

---

> > > ### Author Response · Authors · 2025-08-02
> > >
> > > **Task: DROP**
> > >
> > > | Model        |    STFT $\downarrow$     |   MR-STFT $\downarrow$   |   MR-MEL $\downarrow$    |      SI-SDR $\uparrow$      |      SI-SNR $\uparrow$      |
> > > |:-------------|:------------------------:|:------------------------:|:------------------------:|:---------------------------:|:---------------------------:|
> > > | AudioEditor  |     $6.59 \pm 6.63$      |     $6.58 \pm 6.69$      |     $6.37 \pm 2.67$      |      $-51.24 \pm 9.26$      |      $-51.20 \pm 9.28$      |
> > > | ZETA_50      | $\mathbf{5.58 \pm 7.22}$ | $\mathbf{5.57 \pm 7.63}$ |     $5.51 \pm 2.17$      |     $-20.21 \pm 16.19$      |     $-20.38 \pm 16.56$      |
> > > | ZETA_75      |     $6.27 \pm 8.88$      |     $6.29 \pm 9.41$      |     $5.93 \pm 2.31$      |     $-23.28 \pm 18.21$      |     $-23.06 \pm 17.93$      |
> > > | SAO-Instruct |     $6.12 \pm 11.63$     |     $6.06 \pm 11.68$     | $\mathbf{4.44 \pm 3.08}$ | $\mathbf{-13.77 \pm 16.86}$ | $\mathbf{-13.69 \pm 16.75}$ |
> > >
> > >
> > > **Task: SWAP**
> > >
> > > | Model        |    STFT $\downarrow$     |   MR-STFT $\downarrow$   |   MR-MEL $\downarrow$    |      SI-SDR $\uparrow$      |      SI-SNR $\uparrow$      |
> > > |:-------------|:------------------------:|:------------------------:|:------------------------:|:---------------------------:|:---------------------------:|
> > > | AudioEditor  |     $6.22 \pm 1.13$      |     $6.07 \pm 1.11$      |     $11.78 \pm 2.71$     |     $-60.70 \pm 13.89$      |     $-59.94 \pm 13.77$      |
> > > | ZETA_50      |     $4.39 \pm 1.34$      |     $4.29 \pm 1.31$      |     $6.70 \pm 2.30$      |     $-57.52 \pm 10.98$      |     $-58.01 \pm 10.28$      |
> > > | ZETA_75      |     $4.56 \pm 1.58$      |     $4.46 \pm 1.56$      | $\mathbf{6.45 \pm 1.72}$ |      $-56.45 \pm 9.86$      |      $-56.53 \pm 9.65$      |
> > > | SAO-Instruct | $\mathbf{4.38 \pm 1.05}$ | $\mathbf{4.25 \pm 1.04}$ |     $7.32 \pm 2.17$      | $\mathbf{-54.19 \pm 11.35}$ | $\mathbf{-54.30 \pm 11.36}$ |
> > >
> > >
> > > **Task: LOOP**
> > >
> > > | Model        |    STFT $\downarrow$     |   MR-STFT $\downarrow$   |   MR-MEL $\downarrow$    |      SI-SDR $\uparrow$      |      SI-SNR $\uparrow$      |
> > > |:-------------|:------------------------:|:------------------------:|:------------------------:|:---------------------------:|:---------------------------:|
> > > | AudioEditor  |     $6.34 \pm 2.23$      |     $6.21 \pm 2.19$      |     $10.39 \pm 3.90$     |     $-55.31 \pm 12.19$      |     $-55.19 \pm 11.99$      |
> > > | ZETA_50      |     $4.70 \pm 1.85$      |     $4.55 \pm 1.83$      |     $8.01 \pm 3.41$      |     $-18.13 \pm 12.79$      |     $-18.10 \pm 12.75$      |
> > > | ZETA_75      |     $5.11 \pm 2.29$      |     $4.95 \pm 2.27$      |     $8.26 \pm 3.34$      |     $-19.61 \pm 14.10$      |     $-19.58 \pm 14.28$      |
> > > | SAO-Instruct | $\mathbf{2.01 \pm 0.58}$ | $\mathbf{1.94 \pm 0.58}$ | $\mathbf{2.36 \pm 0.93}$ | $\mathbf{-11.91 \pm 13.78}$ | $\mathbf{-11.89 \pm 13.76}$ |
> > >
> > >
> > > **Task: PITCH**
> > >
> > > | Model        |    STFT $\downarrow$     |   MR-STFT $\downarrow$   |   MR-MEL $\downarrow$    |      SI-SDR $\uparrow$      |      SI-SNR $\uparrow$      |
> > > |:-------------|:------------------------:|:------------------------:|:------------------------:|:---------------------------:|:---------------------------:|
> > > | AudioEditor  |     $5.73 \pm 4.04$      |     $5.73 \pm 4.20$      |     $5.79 \pm 1.97$      |     $-51.73 \pm 11.31$      |     $-51.40 \pm 10.97$      |
> > > | ZETA_50      |     $5.44 \pm 7.75$      |     $5.42 \pm 8.32$      |     $4.78 \pm 1.55$      |     $-46.78 \pm 13.71$      |     $-46.79 \pm 13.79$      |
> > > | ZETA_75      |     $5.95 \pm 8.82$      |     $5.94 \pm 9.44$      |     $5.08 \pm 1.85$      |     $-48.66 \pm 14.11$      |     $-48.59 \pm 14.21$      |
> > > | SAO-Instruct | $\mathbf{2.63 \pm 0.70}$ | $\mathbf{2.49 \pm 0.71}$ | $\mathbf{3.11 \pm 0.96}$ | $\mathbf{-41.39 \pm 14.74}$ | $\mathbf{-41.31 \pm 14.42}$ |
> > >
> > >
> > > **Task: SPEED**
> > >
> > > | Model        |    STFT $\downarrow$     |   MR-STFT $\downarrow$   |   MR-MEL $\downarrow$    |      SI-SDR $\uparrow$      |      SI-SNR $\uparrow$      |
> > > |:-------------|:------------------------:|:------------------------:|:------------------------:|:---------------------------:|:---------------------------:|
> > > | AudioEditor  |     $6.49 \pm 4.85$      |     $6.45 \pm 4.98$      |     $7.02 \pm 3.03$      |     $-50.96 \pm 13.55$      |     $-50.60 \pm 12.52$      |
> > > | ZETA_50      |     $4.94 \pm 2.82$      |     $4.83 \pm 2.83$      |     $5.84 \pm 2.44$      |     $-45.93 \pm 13.81$      |     $-46.75 \pm 13.66$      |
> > > | ZETA_75      |     $5.33 \pm 2.97$      |     $5.23 \pm 2.99$      |     $6.22 \pm 2.54$      |     $-47.28 \pm 12.97$      |     $-47.39 \pm 12.85$      |
> > > | SAO-Instruct | $\mathbf{3.53 \pm 1.23}$ | $\mathbf{3.43 \pm 1.26}$ | $\mathbf{5.13 \pm 1.37}$ | $\mathbf{-45.12 \pm 13.79}$ | $\mathbf{-46.15 \pm 13.00}$ |

---

> > > > ### Author Response · Authors · 2025-08-02
> > > >
> > > > **Task: LOW_PASS**
> > > >
> > > > | Model        |    STFT $\downarrow$     |   MR-STFT $\downarrow$   |   MR-MEL $\downarrow$    |     SI-SDR $\uparrow$      |     SI-SNR $\uparrow$      |
> > > > |:-------------|:------------------------:|:------------------------:|:------------------------:|:--------------------------:|:--------------------------:|
> > > > | AudioEditor  |     $4.94 \pm 2.17$      |     $4.92 \pm 2.16$      |     $5.92 \pm 1.90$      |     $-53.40 \pm 10.43$     |     $-53.99 \pm 10.64$     |
> > > > | ZETA_50      |     $3.41 \pm 1.27$      |     $3.32 \pm 1.27$      |     $4.87 \pm 1.76$      |     $-16.07 \pm 13.30$     |     $-15.99 \pm 13.30$     |
> > > > | ZETA_75      |     $3.79 \pm 1.54$      |     $3.70 \pm 1.48$      |     $5.28 \pm 1.95$      |     $-17.76 \pm 15.49$     |     $-17.78 \pm 15.47$     |
> > > > | SAO-Instruct | $\mathbf{1.63 \pm 0.45}$ | $\mathbf{1.56 \pm 0.47}$ | $\mathbf{1.89 \pm 0.59}$ | $\mathbf{-2.26 \pm 10.95}$ | $\mathbf{-2.28 \pm 10.95}$ |
> > > >
> > > >
> > > > **Task: HIGH_PASS**
> > > >
> > > > | Model        |    STFT $\downarrow$     |   MR-STFT $\downarrow$   |   MR-MEL $\downarrow$    |      SI-SDR $\uparrow$      |      SI-SNR $\uparrow$      |
> > > > |:-------------|:------------------------:|:------------------------:|:------------------------:|:---------------------------:|:---------------------------:|
> > > > | AudioEditor  |     $6.95 \pm 7.75$      |     $6.94 \pm 7.74$      |     $6.52 \pm 1.46$      |     $-62.28 \pm 12.08$      |     $-62.25 \pm 12.08$      |
> > > > | ZETA_50      |     $4.89 \pm 3.42$      |     $4.82 \pm 3.43$      |     $5.15 \pm 1.77$      |     $-31.05 \pm 12.92$      |     $-30.98 \pm 13.01$      |
> > > > | ZETA_75      |     $5.82 \pm 5.53$      |     $5.71 \pm 5.46$      |     $5.65 \pm 2.12$      |     $-34.64 \pm 15.85$      |     $-34.23 \pm 15.52$      |
> > > > | SAO-Instruct | $\mathbf{1.75 \pm 0.38}$ | $\mathbf{1.68 \pm 0.40}$ | $\mathbf{2.08 \pm 0.59}$ | $\mathbf{-28.48 \pm 12.66}$ | $\mathbf{-28.48 \pm 12.66}$ |
> > > >
> > > >
> > > > **Task: INPAINT**
> > > >
> > > > | Model        |    STFT $\downarrow$     |   MR-STFT $\downarrow$   |   MR-MEL $\downarrow$    |      SI-SDR $\uparrow$      |      SI-SNR $\uparrow$      |
> > > > |:-------------|:------------------------:|:------------------------:|:------------------------:|:---------------------------:|:---------------------------:|
> > > > | AudioEditor  |     $6.50 \pm 3.80$      |     $6.43 \pm 3.91$      |     $9.81 \pm 4.40$      |     $-54.40 \pm 12.39$      |     $-54.71 \pm 11.92$      |
> > > > | ZETA_50      |     $4.61 \pm 3.45$      |     $4.54 \pm 3.44$      |     $5.33 \pm 2.13$      |     $-22.34 \pm 17.05$      |     $-22.51 \pm 17.09$      |
> > > > | ZETA_75      |     $5.62 \pm 6.32$      |     $5.53 \pm 6.28$      |     $5.13 \pm 1.74$      |     $-23.49 \pm 16.82$      |     $-24.21 \pm 17.06$      |
> > > > | SAO-Instruct | $\mathbf{2.49 \pm 1.00}$ | $\mathbf{2.42 \pm 0.99}$ | $\mathbf{3.31 \pm 2.24}$ | $\mathbf{-13.98 \pm 13.20}$ | $\mathbf{-14.61 \pm 12.85}$ |
> > > >
> > > >
> > > > **Task: SUPER_RES**
> > > >
> > > > | Model        |    STFT $\downarrow$     |   MR-STFT $\downarrow$   |   MR-MEL $\downarrow$    |     SI-SDR $\uparrow$     |     SI-SNR $\uparrow$     |
> > > > |:-------------|:------------------------:|:------------------------:|:------------------------:|:-------------------------:|:-------------------------:|
> > > > | AudioEditor  |     $5.56 \pm 2.66$      |     $5.53 \pm 2.66$      |     $6.66 \pm 1.75$      |    $-52.83 \pm 12.23$     |    $-52.93 \pm 12.31$     |
> > > > | ZETA_50      |     $4.51 \pm 3.02$      |     $4.43 \pm 3.02$      |     $4.68 \pm 1.63$      |    $-14.19 \pm 11.74$     |    $-14.18 \pm 11.75$     |
> > > > | ZETA_75      |     $4.75 \pm 3.47$      |     $4.67 \pm 3.47$      |     $4.96 \pm 1.70$      |    $-16.97 \pm 14.71$     |    $-16.96 \pm 14.74$     |
> > > > | SAO-Instruct | $\mathbf{2.16 \pm 0.71}$ | $\mathbf{2.10 \pm 0.71}$ | $\mathbf{2.36 \pm 0.81}$ | $\mathbf{-1.68 \pm 9.70}$ | $\mathbf{-1.67 \pm 9.69}$ |
> > > >
> > > >
> > > > **Task: DENOISE**
> > > >
> > > > | Model        |    STFT $\downarrow$     |   MR-STFT $\downarrow$   |   MR-MEL $\downarrow$    |     SI-SDR $\uparrow$      |     SI-SNR $\uparrow$      |
> > > > |:-------------|:------------------------:|:------------------------:|:------------------------:|:--------------------------:|:--------------------------:|
> > > > | AudioEditor  |     $5.30 \pm 6.38$      |     $5.29 \pm 6.38$      |     $5.00 \pm 1.66$      |     $-52.65 \pm 10.11$     |     $-52.79 \pm 10.24$     |
> > > > | ZETA_50      |     $4.79 \pm 2.17$      |     $4.70 \pm 2.17$      |     $5.28 \pm 2.02$      |     $-15.31 \pm 14.27$     |     $-15.16 \pm 14.28$     |
> > > > | ZETA_75      |     $4.91 \pm 2.70$      |     $4.83 \pm 2.70$      |     $6.10 \pm 2.59$      |     $-18.37 \pm 16.37$     |     $-18.34 \pm 16.62$     |
> > > > | SAO-Instruct | $\mathbf{2.36 \pm 0.50}$ | $\mathbf{2.30 \pm 0.48}$ | $\mathbf{2.74 \pm 0.56}$ | $\mathbf{-3.39 \pm 11.03}$ | $\mathbf{-3.50 \pm 11.02}$ |

---

> > > ### Comment · Reviewer_AzCo · 2025-08-03
> > >
> > > I will upgrade my score from 2 to 4 as the authors provided separate evaluation results for 10 types of editing operations.

---

> > > > ### Author Response · Authors · 2025-08-03
> > > >
> > > > We thank the reviewer for increasing their score. It is greatly appreciated.

---

### Official Review · Reviewer_HFny · 2025-07-03

**Clarity:** 3
**Significance:** 3
**Originality:** 3
**Rating:** 5
**Confidence:** 4

**Summary:**

The paper presents SAO-Instruct, which fine-tunes the previous stable audio open into an audio editing model. The main efforts of this are in the data synthesis, which aims to generate the (input audio, instruction, output audio) triplets. The author presents 3 different data synthesis methods, aka, prompt-to-prompt, DDPM inversion, and manual edition.

The evaluation is on AudioCaps, with subjective and objective evaluation results.
SAO-Instruct t shows similar generation quality to previous baseline models, while claiming better relevance and faithfulness.

**Questions:**

As above

**Ethical Concerns:**

["NO or VERY MINOR ethics concerns only"]

**Final Justification:**

The initial manuscript is already good enough, which provides a good implementation for free-form audio generation. My concern/questions are mostly based on the data and implementation details, which have already been well addressed during rebuttal stage. I have no more concerns.

**Limitations:**

Some recent peer-reviewed works are missing from the related work. Especially, Fuggato has claimed the free-form instruction-following and editing.

https://openreview.net/forum?id=B2Fqu7Y2cd
https://arxiv.org/abs/2412.19351

**Paper Formatting Concerns:**

No concern

**Quality:**

3

**Strengths And Weaknesses:**

Strengths

(1) The work is well-motivated, with clear writing. Although most techniques are inherited from image generation/editing, it has been well adapted in the audio generation scenarios.

(2) The authors present 3 different synthesis techniques, which cover fully-synthetic, partially-synthetic, and manual edits, which makes the exploration comprehensive.

(3) The evaluation contains both subjective and objective aspects, with multiple aspects and metrics.

Weakness:
(1) As the main efforts of this work should be considered in data synthesis, I would recommend the authors to provide more insights into this process. Specifically, for each of the three data synthesis methods, how do you monitor the intermediate results quality, what are the statistical numbers along this way, and how is data quality control achieved and impact the overall results. Since the model is only fine-tuned on 50k samples, I would expect the authors to run more ablation studies to dig out what factors during the data synthesis would have the most impact on the final performance.

(2) The author claims their model can follow the free-form instructions. However, when using the manual edition method alone (note this is implemented with a closed set of operations), it shows close performance with the other two methods. That's to say, under the evaluation paradigm provided by the authors, the ability to follow free-form instruction is not well justified.

(3) When claiming "achieving both accurate edits and faithfulness" in Table 2, do the authors want to mention the MOS scores like Table 3?

(4) This is optional, but is recommended: can the authors provide some generated samples for demonstration?

---

> ### Author Rebuttal · Authors · 2025-07-30
>
> We thank the reviewer for their valuable insights and address their feedback in the following:
>
> > (1) As the main efforts of this work should be considered in data synthesis, I would recommend the authors to provide more insights into this process. Specifically, for each of the three data synthesis methods, how do you monitor the intermediate results quality, what are the statistical numbers along this way, and how is data quality control achieved and impact the overall results.
>
> We agree that the quality of our generated data is critical for our approach. This is why we use a two-stage pipeline for the fully synthetic sample generation using Prompt-to-Prompt. Candidates (seed/CFG values) are first filtered using a perceptual Gemini check and CLAP similarity before the final sample is generated. Because intermediate statistics can be misleading across very different free-form edits (e.g., a subtle “add a light echo” vs. a structural “remove people talking”), we relied on subjective listening during development to set suitable ranges for the parameters we optimize (e.g., attention injection strength). To have a proxy for human judgement of generated samples at scale, we defined an objective function used during Bayesian Optimization in Prompt-to-Prompt and DDPM Inversion that combines several metrics. We calibrated the metric weights in a small scale listening study where listeners compared samples picked by the objective function initialized with different weightings. Finally, we selected the top-ranked weight initialization based on an ELO rating as described in Section 3.2 (Objective Function). We clarified our reliance on subjective listening for design decisions and the selection of parameter ranges in the revised manuscript.
>
> > Since the model is only fine-tuned on 50k samples, I would expect the authors to run more ablation studies to dig out what factors during the data synthesis would have the most impact on the final performance.
>
> We report an ablation study isolating the contribution of each proposed data generation method (Prompt-to-Prompt, DDPM inversion, and manual edits) and their combination, showing complementary gains (quality and edit accuracy) when all are used together. During development we conducted internal listening tests that informed our in-method design decisions (e.g., the perceptual Gemini filter during Prompt-to-Prompt). Exhaustive ablations for each data generation method, especially for the fully or semi-synthetic data generation methods, would require a significantly higher compute budget. Therefore, our ablation study focuses on comparing the three methods and their combination.
>
> > (2) The author claims their model can follow the free-form instructions. However, when using the manual edition method alone (note this is implemented with a closed set of operations), it shows close performance with the other two methods. That's to say, under the evaluation paradigm provided by the authors, the ability to follow free-form instruction is not well justified.
>
> We thank the reviewer for this observation. The model fine-tuned on manual edits scores well on metrics computed against the original audio indicating high content preservation and quality. However, it underperforms on metrics computed against the caption-conditioned regenerated reference, which indicates weaker edit instruction adherence. This trade-off is also evident in the MOS for the ablation study that we provide in this response. While the manual-only model can still follow some free-form edit instructions, mainly due to the Stable Audio Open pretraining and our diverse edit instruction generation, it cannot follow the same broad range of free-form edits covered by the Prompt-to-Prompt and DDPM pipelines (e.g., “make the car drive on gravel instead”).
>
> > (3) When claiming "achieving both accurate edits and faithfulness" in Table 2, do the authors want to mention the MOS scores like Table 3?
>
> We provide the MOS for the ablation in the following table:
>
> | Training Dataset     | Quality $\uparrow$ | Relevance $\uparrow$ | Faithfulness $\uparrow$ |
> | --------------------- | ---------------------: | --------------------------: | --------------------------: |
> | Prompt-to-Prompt | $3.32 \pm 0.79$     | $3.92 \pm 0.80$           | $3.48 \pm 0.83$             |
> | DDPM Inversion     | $3.42 \pm 0.84$     | $2.80 \pm 1.16$           | $3.90 \pm 0.68$             |
> | Manual Edits          | $3.78 \pm 0.95$     | $2.00 \pm 1.01$           | $4.48 \pm 0.61$             |
> | Combined              | $3.52 \pm 0.81$     | $3.76 \pm 1.10$           | $3.56 \pm 0.76$             |
>
> Fine-tuning on the Prompt-to-Prompt dataset achieves the highest edit relevance but at a slight cost in quality and faithfulness to the input audio (it can over-edit and introduce perceptual artifacts). DDPM inversion improves quality and faithfulness, but tends to follow instructions less precisely. Using only manual edits achieves the best faithfulness and quality but suffers from low edit relevance due to the closed operation set. The combined setting balances the advantages of these methods and achieves high relevance while maintaining strong quality and faithfulness to the input audio characteristics.
>
> > (4) This is optional, but is recommended: can the authors provide some generated samples for demonstration?
>
> The manuscript on page 2 (Line 57) includes a link to our online demo page with generated samples. We will make it more prominent in the revised manuscript.
>
> > Some recent peer-reviewed works are missing from the related work. Especially, Fuggato has claimed the free-form instruction-following and editing.
>
> We thank the reviewer for noting this. We will expand the related work section in our manuscript to include these recent works and briefly clarify how our setting and contributions differ.
>
> Based on our feedback and proposed improvements, would the reviewer be willing to reconsider their score?

---

> > ### Comment · Reviewer_HFny · 2025-08-08
> >
> > I appreciate the detailed reply from the authors. I would change my score to 5

---

### Official Review · Reviewer_t9mM · 2025-07-07

**Clarity:** 3
**Significance:** 4
**Originality:** 3
**Rating:** 5
**Confidence:** 4

**Summary:**

This paper presents **SAO-Instruct**, a model fine-tuned from the Stable Audio Open model on triplets (audio, text instruction, target audio) to enable free-form, instruction-based audio editing. The approach uses a supervised training pipeline and proposes a scalable method for generating synthetic instruction-editing data using a combination of prompt-to-prompt, DDPM inversion, and manual editing methods. For prompt-to-prompt and DDPM inversion, an LLM is used to generate paired instructions and output captions, which are then filtered using a CLAP-based objective function. For manual editing, a set of twelve editing tasks inspired by AUDIT is used. Experiments on datasets constructed from AudioSet, Freesound, and AudioSet-SL (50K for ablations and 150K for final training) demonstrate the effectiveness of SAO-Instruct, which significantly outperforms baselines in subjective evaluations.

**Questions:**

- The authors mention fine-tuning SAO on AudioCaps to improve prompt adherence for general audio. It may also be useful to report metrics such as KLD, IS, and FAD, which are commonly used in the literature.
- Table 3 suggests that objective scores do not align with subjective evaluations. Have you considered using directional CLAP, similar to your filtering setup, to better capture relevance? For quality, have the authors looked at perceptual metrics like those proposed in [3,4]?
- Lines 259–260: "During inference, we encode the input audio into the latent space of the diffusion model and add Gaussian noise..."
Why is the noised latent used during inference, this seems to be different from training time?  Did the authors compare this with sampling from noise directly? Adding those results will be useful.

**Ethical Concerns:**

["NO or VERY MINOR ethics concerns only"]

**Final Justification:**

I have read the review and rebuttals and decided to keep my original rating

**Limitations:**

Yes

**Quality:**

3

**Strengths And Weaknesses:**

**Strengths**:

- The paper is well written and generally easy to follow.
- SAO-Instruct significantly extends the coverage of text instructions for audio editing and outperforms baselines, especially on relevance and faithfulness. The inference speedups are also quite substantial.
- The proposed data generation pipeline for instruction-based audio editing is scalable, and the released code for sample generation and training would help with reproducibility.

**Weaknesses**:
- The dataset ablations are not very convincing, particularly because in Table 3 subjective and objective metrics do not align well. It would be more useful if other metrics like in [3, 4] were included that align better with subjective judgments or evaluations
- The reliability of the objective metrics is unclear. The authors suggest using both the original and generated audio as references; however, for complex edits, measuring KL or LSD against the original audio may not be meaningful, as the event distribution can change significantly.
- Several choices in the data generation pipeline are not well-motivated, and discussion is missing:
    - Prompt-to-prompt sample generation uses 100 steps (line 200) and DDPM uses 70 (line 218).  A discussion of this design choice would help.
    - Lines 247-250 describe the data composition for Prompt-to-prompt, DDPM, and manual edits without much discussion around the choices or sampling strategy if any.
    - Line 286: "We evaluate on 1k 10-second samples from the AudioCaps test subset..." — the composition of the test set is unclear. Given that manual and prompt-to-prompt edits may differ in distribution, this could influence conclusions. Would the authors consider releasing the test set?

*Minor Comments / Suggestions**
- Lines 68-72: "One downside of using mel spectrograms ... continuous latent representation"
  While it is true that Mel-Spectrogram based vocoders can suffer from missing phase information,  VAE-based continuous representations can also introduce artifacts. BigVGAN-based vocoders perform well in practice, so this statement could be rephrased to present a more balanced view.
- Besides the prompt-to-prompt and zero-shot editing approaches discussed in Section 2.2, diffusion models [1, 2] trained with masked in-filling support editing via masked audio + target text. These models could be used to generate instruction fine-tuning data for datasets like AudioSet-Strongly Labelled, which provides event timestamps. Including this perspective may be useful for a general audience.

**References**:
[1] Unified Audio Generation with Natural Language Prompts. Arxiv 2023

[2] Fugatto 1: Foundational generative audio transformer opus 1. ICLR 2025

[3] Meta Audiobox Aesthetics: Unified Automatic Quality Assessment for Speech, Music, and Sounda. Arxiv 2023

[4] https://github.com/wavlab-speech/versa

---

> ### Author Rebuttal · Authors · 2025-07-30
>
> We thank the reviewer for their valuable insights and address their feedback in the following:
>
> > The reliability of the objective metrics is unclear. The authors suggest using both the original and generated audio as references; however, for complex edits, measuring KL or LSD against the original audio may not be meaningful, as the event distribution can change significantly.
>
> We thank the reviewer for this remark. The main challenge we face during evaluation is the lack of ground-truth edited general audio pairs. Following AudioEditor [1], we use two complementary reference proxies: (i) scores against the original audio clips to measure quality and content preservation to guard against over-editing, and (ii) scores against a regenerated sample conditioned on the target caption to approximate edit accuracy. These scores reflect similarity to their respective references rather than absolute correctness and should be evaluated jointly to assess the model’s performance. As future work, datasets exploring edited general audio pairs would enable a more direct evaluation of edit accuracy.
>
> > Prompt-to-prompt sample generation uses 100 steps (line 200) and DDPM uses 70 (line 218).
>
> We use different step counts because these methods have different goals and compute profiles. For Prompt-to-Prompt, we found that generating the final selected pair with 100 denoising steps results in high-quality synthetic samples. The DDPM inversion approach using ZETA consists of two stages: (i) the input audio is inverted to an initial latent up to T_start, (ii) this latent is conditioned on the target caption and denoised for 70 steps. As this two-stage process results in a longer inference time compared to Prompt-to-Prompt, 70 denoising steps gave a good quality-compute trade-off for ZETA. We clarified these choices in the revised manuscript.
>
> > Lines 247-250 describe the data composition for Prompt-to-prompt, DDPM, and manual edits without much discussion around the choices or sampling strategy if any.
>
> We use a random subset of AudioSet‑SL and Freesound, plus the full AudioCaps train split. We then compute the number of audible elements per caption using the procedure in Section 3.1 (Prompt Generation) and discard any sample with >2 elements, as mentioned in Appendix A.
>
> > "We evaluate on 1k 10-second samples from the AudioCaps test subset..." — the composition of the test set is unclear. Given that manual and prompt-to-prompt edits may differ in distribution, this could influence conclusions. Would the authors consider releasing the test set?
>
> We thank the reviewer for raising this point. We are considering releasing the prompt test set for reproducibility which contains input captions from AudioCaps. However, AudioCaps states that their dataset may only be used for academic purposes.
>
> > Minor Comments / Suggestions
>
> In regards to the minor comments and suggestions: We agree that mel-spectrogram based vocoders have achieved great results and have revised Section 2.1 to provide a more balanced view. We also expanded Section 2.2 to mention current work in masked in-filling editing, including Fugatto.
>
> > The authors mention fine-tuning SAO on AudioCaps to improve prompt adherence for general audio. It may also be useful to report metrics such as KLD, IS, and FAD, which are commonly used in the literature.
>
> We thank the reviewer for this suggestion. In the current manuscript, we compared SAO with its version fine-tuned on the AudioCaps training subset using CLAP similarity. We now also report FD, KL, and IS to better motivate our choice to fine-tune SAO, with all metrics evaluated on the same AudioCaps test subset for consistency. We report FD computed with PANNs features rather than FAD with VGGish, as the PANNs classifier provided more reliable results in our experiments and in prior work [2].
>
> |  Model                                  | FD $\downarrow$ |  KL $\downarrow$ |       IS $\uparrow$ |  CLAP $\uparrow$ |
> | -------------------------------- | --------------------: | --------------------: | --------------------: | --------------------: |
> | SAO                                      |                 $41.64$ |                    $2.19$ |   $8.56 \pm 0.47$ |    $0.26 \pm 0.14$ |
> | SAO AudioCaps Fine-tuned |                 $20.85$ |                    $1.42$ | $10.05 \pm 0.51$ |    $0.46 \pm 0.11$ |
>
> > Lines 259–260: "During inference, we encode the input audio into the latent space of the diffusion model and add Gaussian noise..." Why is the noised latent used during inference, this seems to be different from training time? Did the authors compare this with sampling from noise directly? Adding those results will be useful.
>
> We have added an ablation table comparing two inference modes for SAO-Instruct: sampling from pure-noise versus sampling from the Gaussian-noised latent of the encoded input audio. Metrics with * are computed against the regenerated reference, while the others use the original audio. Starting from the noised latent substantially improves metrics computed against the original audio clips, which indicates better preservation of the input audio's characteristics. It also achieves a higher CLAP score and lower FD/KL relative to the target caption-conditioned regenerated reference, showing accurate instruction following. Overall, sampling from the noised latent preserves more input audio characteristics while still providing enough flexibility to perform the required edits. We will include these results in the Appendix of the revised manuscript.
>
> | Reference                                             | FD $\downarrow$ | LSD $\downarrow$ | KL $\downarrow$ | FD* $\downarrow$ | LSD* $\downarrow$ | KL* $\downarrow$ | IS $\uparrow$ | CLAP $\uparrow$ |
> | ------------------------------------------- | :--------- | :-------- | :------ | :--------- | :------- | :------- | :----------------- | :------------------ |
> | SAO‑Instruct (without encoded audio) | $29.73$ | $2.23$ | $2.59$ | $19.17$ | $2.61$ | $2.27$ | $8.10 \pm 0.69$ | $0.32 \pm 0.15$ |
> | SAO‑Instruct (with encoded audio)      | $18.38$ | $1.36$ | $0.93$ | $18.97$ | $2.72$ | $1.76$ | $7.59 \pm 1.00$ | $0.38 \pm 0.14$ |
>
>
> [1] Jia, Yuhang, et al. "AudioEditor: A training-free diffusion-based audio editing framework." ICASSP 2025-2025 IEEE International Conference on Acoustics, Speech and Signal Processing (ICASSP). IEEE, 2025.
>
> [2] Liu, Haohe, et al. "AudioLDM: Text-to-audio generation with latent diffusion models." Proceedings of the 40th International Conference on Machine Learning. 2023.

---

> > ### Comment · Reviewer_t9mM · 2025-08-05
> > **Further question**
> >
> > Thank you for taking time to answer the questions. Could the authors also comment on the missing question: "Table 3 suggests that objective scores do not align with subjective evaluations. Have you considered using directional CLAP, similar to your filtering setup, to better capture relevance? For quality, have the authors looked at perceptual metrics like those proposed in [3,4]?"

---

> > > ### Author Response · Authors · 2025-08-06
> > >
> > > > Table 3 suggests that objective scores do not align with subjective evaluations. Have you considered using directional CLAP, similar to your filtering setup, to better capture relevance?
> > >
> > > We apologize for missing this question in our initial response, and we appreciate the opportunity to clarify. Directional-CLAP rewards the direction of change but is largely insensitive to the magnitude. Consider the instruction “there should be fireworks”. If the model inserts a single distant firework explosion, the audio embedding shifts towards the correct direction, resulting in a relatively high directional CLAP score. However, listeners would judge the edit as insufficient, yielding low Relevance. If the model adds frequent, loud fireworks that dominate the audio clip, the embedding moves again in the correct direction, resulting in a high directional CLAP score. Listeners may give this edit low Relevance as it does not match the instruction’s intent (“there should be fireworks” does not imply that they should greatly dominate the audio clip). In our preliminary experiments, the directional CLAP score did not reliably follow human judgment, so we did not use it as an evaluation metric.
> > >
> > > As the reviewer notes, we do use directional CLAP in our filtering pipeline, where we combine it as one part of a composite signal. The multi-scale mel loss and the CLAP similarity between input and output audio ensures that the edited audio remains faithful to the input audio, while the CLAP direction and CLAP similarity between edited audio and target caption optimize for relevant edits. In this context, we use directional CLAP as a guidance signal, not as a standalone evaluator.
> > >
> > > > For quality, have the authors looked at perceptual metrics like those proposed in [3,4]?
> > >
> > > We thank the reviewer for this suggestion. We will update our revised manuscript to include metrics from AudioBox Aesthetic that are relevant to our evaluation. Production Quality (PQ) measures technical aspects of audio quality, while Production Complexity (PC) focuses on the complexity of the audio scene [1]. The following table shows the results for the models. We observe that SAO-Instruct slightly underperforms ZETA in production quality, while slightly outperforming ZETA in production complexity. These results seem to reflect the findings from our subjective listening study.
> > >
> > > | Model            | Production Quality (PQ) $\uparrow$ | Production Complexity (PC) $\uparrow$ |
> > > |:----------------|:----------------------------------------:|:----------------------------------------------:|
> > > | AudioEditor   |   $5.38 \pm 0.91$                             |   $3.02 \pm 0.72$                                     |
> > > | ZETA_50        |  $\mathbf{6.06 \pm 0.94}$               |  $2.76 \pm 0.69$                                     |
> > > | ZETA_75        |  $6.04 \pm 0.91$                              |  $2.73 \pm 0.66$                                     |
> > > | SAO-Instruct |  $5.61 \pm 0.89$                              |  $\mathbf{3.23 \pm 0.78}$                        |
> > >
> > >
> > > [1] Tjandra, Andros, et al. "Meta Audiobox Aesthetics: Unified automatic quality assessment for speech, music, and sound." arXiv preprint arXiv:2502.05139 (2025).

---

### Decision · Program_Chairs · 2025-09-17

**Decision:**

Accept (poster)

**Comment:**

Audio editing is a relatively underexplored area where most focus has been on image editing. Although some of the techniques are derived from image-based generative models, the reviewers still appreciated the effort in translating those to the audio domain. There are positive results compared to sensitive baselines. Most of the questions by reviewers were aimed at the data generation process as the paper also claims to provide a valuable data resource that is generated partially through synthetic data generation. Reviewers were mostly satisfied and provided positive scores overall. The AC reviewed the paper and agrees with the reviewer consensus.